# Practical and Consistent Estimation of $f$-Divergences

**Paul K. Rubenstein**[*]
Max Planck Institute for Intelligent Systems, Tübingen
& Machine Learning Group, University of Cambridge
paul.rubenstein@tuebingen.mpg.de

**Olivier Bousquet, Josip Djolonga, Carlos Riquelme, Ilya Tolstikhin**
Google Research, Brain Team, Zürich
{obousquet, josipd, rikel, tolstikhin}@google.com

## Abstract

The estimation of an $f$-divergence between two probability distributions based on samples is a fundamental problem in statistics and machine learning. Most works study this problem under very weak assumptions, in which case it is provably hard. We consider the case of stronger structural assumptions that are commonly satisfied in modern machine learning, including representation learning and generative modelling with autoencoder architectures. Under these assumptions we propose and study an estimator that can be easily implemented, works well in high dimensions, and enjoys faster rates of convergence. We verify the behavior of our estimator empirically in both synthetic and real-data experiments, and discuss its direct implications for total correlation, entropy, and mutual information estimation.

## 1 Introduction and related literature

The estimation and minimization of divergences between probability distributions based on samples are fundamental problems of machine learning. For example, maximum likelihood learning can be viewed as minimizing the Kullback-Leibler divergence $\mathrm{KL}(P_{\mathrm{data}}\|P_{\mathrm{model}})$ with respect to the model parameters. More generally, generative modelling—most famously Variational Autoencoders and Generative Adversarial Networks [21, 12]—can be viewed as minimizing a divergence $D(P_{\mathrm{data}}\|P_{\mathrm{model}})$ where $P_{\mathrm{model}}$ may be intractable. In variational inference, an intractable posterior $p(z|x)$ is approximated with a tractable distribution $q(z)$ chosen to minimize $\mathrm{KL}\big(q(z)\|p(z|x)\big)$. The mutual information between two variables $I(X, Y)$, core to information theory and Bayesian machine learning, is equivalent to $\mathrm{KL}(P_{X,Y}\|P_X P_Y)$. Independence testing often involves estimating a divergence $D(P_{X,Y}\|P_X P_Y)$, while two-sample testing (does $P = Q$?) involves estimating a divergence $D(P\|Q)$. Additionally, one approach to domain adaptation, in which a classifier is learned on a distribution $P$ but tested on a distinct distribution $Q$, involves learning a feature map $\phi$ such that a divergence $D(\phi_\# P\|\phi_\# Q)$ is minimized, where $\phi_\#$ represents the push-forward operation [3, 11].

In this work we consider the well-known family of $f$-divergences [7, 24] that includes amongst others the KL, Jensen-Shannon (JS), $\chi^2$, and $\alpha$-divergences as well as the Total Variation (TV) and squared Hellinger ($\mathrm{H}^2$) distances, the latter two of which play an important role in the statistics literature [2]. A significant body of work exists studying the estimation of the $f$-divergence $D_f(Q\|P)$ between general probability distributions $Q$ and $P$. While the majority of this focuses on $\alpha$-divergences and closely related Rényi-$\alpha$ divergences [35, 37, 22], many works address specifically the KL-divergence [34, 39] with fewer considering $f$-divergences in full generality [28, 20, 26, 27]. Although the KL-divergence is the most frequently encountered $f$-divergence in the machine learning literature,

---

[*]Part of this work was done during an internship at Google.

in recent years there has been a growing interest in other $f$-divergences [30], in particular in the variational inference community where they have been employed to derive alternative evidence lower bounds [5, 23, 9].

The main challenge in computing $D_f(Q\|P)$ is that it requires knowledge of either the densities of both $Q$ and $P$, or the density ratio $dQ/dP$. In studying this problem, assumptions of differing strength can be made about $P$ and $Q$. In the weakest *agnostic* setting, we may be given only a finite number of i.i.d samples from the distributions without any further knowledge about their densities. As an example of stronger assumptions, both distributions may be mixtures of Gaussians [17, 10], or we may have access to samples from $Q$ and have full knowledge of $P$ [15, 16] as in e.g. model fitting.

Most of the literature on $f$-divergence estimation considers the weaker agnostic setting. The lack of assumptions makes such work widely applicable, but comes at the cost of needing to work around estimation of either the densities of $P$ and $Q$ [37, 22] or the density ratio $dQ/dP$ [28, 20] from samples. Both of these estimation problems are provably hard [2, 28] and suffer rates—the speed at which the error of an estimator decays as a function of the number of samples $N$—of order $N^{-1/d}$ when $P$ and $Q$ are defined over $\mathbb{R}^d$ unless their densities are sufficiently smooth. This is a manifestation of the *curse of dimensionality* and rates of this type are often called *nonparametric*. One could hope to estimate $D_f(P\|Q)$ without explicitly estimating the densities or their ratio and thus avoid suffering nonparametric rates, however a lower bound of the same order $N^{-1/d}$ was recently proved for $\alpha$-divergences [22], a sub-family of $f$-divergences. While some works considering the agnostic setting provide rates for the bias and variance of the proposed estimator [28, 22] or even exponential tail bounds [37], it is more common to only show that the estimators are asymptotically unbiased or consistent without proving specific rates of convergence [39, 35, 20].

Motivated by recent advances in machine learning, we study a setting in which much stronger structural assumptions are made about the distributions. Let $\mathcal{X}$ and $\mathcal{Z}$ be two finite dimensional Euclidean spaces. We estimate the divergence $D_f(Q_Z\|P_Z)$ between two probability distributions $P_Z$ and $Q_Z$, both defined over $\mathcal{Z}$. $P_Z$ has known density $p(z)$, while $Q_Z$ with density $q(z)$ admits the factorization $q(z) := \int_{\mathcal{X}} q(z|x)q(x)dx$ where access to independent samples from the distribution $Q_X$ with unknown density $q(x)$ and full knowledge of the conditional distribution $Q_{Z|X}$ with density $q(z|x)$ are assumed. In most cases $Q_Z$ is intractable due to the integral and so is $D_f(Q_Z\|P_Z)$. As a concrete example, these assumptions are often satisfied in applications of modern unsupervised generative modeling with deep autoencoder architectures, where $\mathcal{X}$ and $\mathcal{Z}$ would be *data* and *latent* spaces, $P_Z$ the *prior*, $Q_X$ the *data distribution*, $Q_{Z|X}$ the *encoder*, and $Q_Z$ the *aggregate posterior*.

Given independent observations $X_1, \ldots, X_N$ from $Q_X$, the finite mixture $\hat{Q}_Z^N := \frac{1}{N}\sum_{i=1}^N Q_{Z|X_i}$ can be used to approximate the continuous mixture $Q_Z$. **Our main contribution** is to approximate the intractable $D_f(Q_Z\|P_Z)$ with $D_f(\hat{Q}_Z^N\|P_Z)$, a quantity that can be estimated to arbitrary precision using Monte-Carlo sampling since both distributions have known densities, and to theoretically study conditions under which this approximation is reasonable. We call $D_f(\hat{Q}_Z^N\|P_Z)$ the Random Mixture (RAM) estimator and derive rates at which it converges to $D_f(Q_Z\|P_Z)$ as $N$ grows. We also provide similar guarantees for RAM-MC—a practical Monte-Carlo based version of RAM. By side-stepping the need to perform density estimation, we obtain *parametric* rates of order $N^{-\gamma}$, where $\gamma$ is independent of the dimension (see Tables 1 and 2), although the constants may still in general show exponential dependence on dimension. This is in contrast to the agnostic setting where *both* nonparametric rates and constants are exponential in dimension.

Our results have immediate implications to existing literature. For the particular case of the KL divergence, a similar approach has been *heuristically* applied independently by several authors for estimating the mutual information [36] and total correlation [6]. Our results provide strong theoretical grounding for these existing methods by showing sufficient conditions for their consistency.

A final piece of related work is [4], which proposes to reduce the gap introduced by Jensen's inequality in the derivation of the classical evidence lower bound (ELBO) by using multiple Monte-Carlo samples from the approximate posterior $Q_{Z|X}$. This is similar in flavour to our approach, but fundamentally different since we use multiple samples from the *data distribution* to reduce a different Jensen gap. To avoid confusion, we note that replacing the "regularizer" term $\mathbb{E}_X[\mathrm{KL}(Q_{Z|X}\|P_Z)]$ of the classical ELBO with expectation of our estimator $\mathbb{E}_{\mathbf{X}^N}[\mathrm{KL}(\hat{Q}_Z^N\|P_Z)]$ results in an upper bound of the classical ELBO (see Proposition 1) but is itself not in general an evidence lower bound:

$$\mathbb{E}_X\left[\mathbb{E}_{Q_{Z|X}}\log p(X|Z) - \mathrm{KL}(Q_{Z|X}\|P_Z)\right] \leq \mathbb{E}_X\left[\mathbb{E}_{Q_{Z|X}}\log p(X|Z)\right] - \mathbb{E}_{\mathbf{X}^N}\left[\mathrm{KL}(\hat{Q}_Z^N\|P_Z)\right].$$

The remainder of the paper is structured as follows. In Section 2 we introduce the RAM and RAM-MC estimators and present our main theoretical results, including rates of convergence for the bias (Theorems 1 and 2) and tail bounds (Theorems 3 and 4). In Section 3 we validate our results in both synthetic and real-data experiments. In Section 4 we discuss further applications of our results. We conclude in Section 5.

## 2   Random mixture estimator and convergence results

In this section we introduce our $f$-divergence estimator, and present theoretical guarantees for it. We assume the existence of probability distributions $P_Z$ and $Q_Z$ defined over $\mathcal{Z}$ with known density $p(z)$ and intractable density $q(z) = \int q(z|x)q(x)dx$ respectively, where $Q_{Z|X}$ is known. $Q_X$ defined over $\mathcal{X}$ is unknown, however we have an i.i.d. sample $\mathbf{X}^N = \{X_1, \dots, X_N\}$ from it. Our ultimate goal is to estimate the intractable $f$-divergence $D_f(Q_Z \| P_Z)$ defined by:

**Definition 1** ($f$-divergence). *Let $f$ be a convex function on $(0, \infty)$ with $f(1) = 0$. The $f$-divergence $D_f$ between distributions $Q_Z$ and $P_Z$ admitting densities $q(z)$ and $p(z)$ respectively is*

$$D_f(Q_Z \| P_Z) := \int f\left(\frac{q(z)}{p(z)}\right) p(z) dz.$$

Many commonly used divergences such as Kullback–Leibler and $\chi^2$ are $f$-divergences. All the divergences considered in this paper together with their corresponding $f$ can be found in Appendix A. Of them, possibly the least well-known in the machine learning literature are $f_\beta$-divergences [32]. These symmetric divergences are continuously parameterized by $\beta \in (0, \infty]$. Special cases include squared-Hellinger ($H^2$) for $\beta = \frac{1}{2}$, Jensen-Shannon (JS) for $\beta = 1$, Total Variation (TV) for $\beta = \infty$.

In our setting $Q_Z$ is intractable and so is $D_f(Q_Z \| P_Z)$. Substituting $Q_Z$ with a sample-based finite mixture $\hat{Q}_Z^N := \frac{1}{N} \sum_{i=1}^N Q_{Z|X_i}$ leads to our proposed **Random Mixture estimator (RAM)**:

$$D_f(\hat{Q}_Z^N \| P_Z) := D_f\left(\frac{1}{N} \sum_{i=1}^N Q_{Z|X_i} \| P_Z\right). \tag{1}$$

Although $\hat{Q}_Z^N$ is a function of $\mathbf{X}^N$ we omit this dependence in notation for brevity. In this section we identify sufficient conditions under which $D_f(\hat{Q}_Z^N \| P_Z)$ is a "good" estimator of $D_f(Q_Z \| P_Z)$. More formally, we establish conditions under which the estimator is asymptotically unbiased, concentrates to its expected value and can be practically estimated using Monte-Carlo sampling.

### 2.1   Convergence rates for the bias of RAM

The following proposition shows that $D_f(\hat{Q}_Z^N \| P_Z)$ upper bounds $D_f(Q_Z \| P_Z)$ in expectation for any finite $N$, and that the upper bound becomes tighter with increasing $N$:

**Proposition 1.** *Let $M \leq N$ be integers. Then*

$$D_f(Q_Z \| P_Z) \leq \mathbb{E}_{\mathbf{X}^N}\left[D_f(\hat{Q}_Z^N \| P_Z)\right] \leq \mathbb{E}_{\mathbf{X}^M}\left[D_f(\hat{Q}_Z^M \| P_Z)\right]. \tag{2}$$

*Proof sketch (full proof in Appendix B.1).* The first inequality follows from Jensen's inequality, using the facts that $f$ is convex and $Q_Z = \mathbb{E}_{\mathbf{X}^N}[\hat{Q}_Z^N]$. The second holds since a sample $\mathbf{X}^M$ can be drawn by sub-sampling (without replacement) $M$ entries of $\mathbf{X}^N$, and by applying Jensen again.   □

As a function of $N$, the expectation is a decreasing sequence that is bounded below. By the monotone convergence theorem, the sequence converges. Theorems 1 and 2 in this section give sufficient conditions under which the expectation of RAM converges to $D_f(Q_Z \| P_Z)$ as $N \to \infty$ for a variety of $f$ and provide rates at which this happens, summarized in Table 1. The two theorems are proved using different techniques and assumptions. These assumptions, along with those of existing methods (see Table 3) are discussed at the end of this section.

**Theorem 1** (Rates of the bias). *If $\mathbb{E}_{X \sim Q_X}\left[\chi^2\left(Q_{Z|X}, Q_Z\right)\right]$ and $\mathrm{KL}\left(Q_Z \| P_Z\right)$ are finite then the bias $\mathbb{E}_{\mathbf{X}^N}\left[D_f(\hat{Q}_Z^N \| P_Z)\right] - D_f\left(Q_Z \| P_Z\right)$ decays with rate as given in the first row of Table 1.*

*Proof sketch (full proof in Appendix B.2).* There are two key steps to the proof. The first is to bound the bias by $\mathbb{E}_{\mathbf{X}^N}\left[D_f(\hat{Q}_Z^N, Q_Z)\right]$. For the KL this is an equality. For $D_{f_\beta}$ this holds because for

Table 1: Rate of bias $\mathbb{E}_{\mathbf{X}^N} D_f\left(\hat{Q}_Z^N \| P_Z\right) - D_f\left(Q_Z \| P_Z\right)$.

| $f$-divergence | KL | TV | $\chi^2$ | H$^2$ | JS | $D_{f_\beta}$ $\frac{1}{2}<\beta<1$ | $D_{f_\beta}$ $1<\beta<\infty$ | $D_{f_\alpha}$ $-1<\alpha<1$ |
|---|---|---|---|---|---|---|---|---|
| Theorem 1 | $N^{-1}$ | $N^{-\frac{1}{2}}$ | - | $N^{-\frac{1}{2}}$ | $N^{-\frac{1}{4}}$ | $N^{-\frac{1}{4}}$ | $N^{-\frac{1}{4}}$ | - |
| Theorem 2 | $N^{-\frac{1}{3}}\log N$ | $N^{-\frac{1}{2}}$ | $N^{-1}$ | $N^{-\frac{1}{5}}$ | $N^{-\frac{1}{3}}\log N$ | $N^{-\frac{1}{3}}$ | $N^{-\frac{1}{2}}$ | $N^{-\frac{\alpha+1}{\alpha+5}}$ |

Table 2: Rate $\psi(N)$ of high probability bounds for $D_f\left(\hat{Q}_Z^N \| P_Z\right)$ (Theorem 3).

| $f$-divergence | KL | TV | $\chi^2$ | H$^2$ | JS | $D_{f_\beta}$ $\frac{1}{2}<\beta<1$ | $D_{f_\beta}$ $1<\beta<\infty$ | $D_{f_\alpha}$ $\frac{1}{3}<\alpha<1$ |
|---|---|---|---|---|---|---|---|---|
| $\psi(N)$ | $N^{-\frac{1}{6}}\log N$ | $N^{-\frac{1}{2}}$ | $N^{-\frac{1}{2}}$ | - | $N^{-\frac{1}{6}}\log N$ | $N^{-\frac{1}{6}}$ | $N^{-\frac{1}{2}}$ | $N^{\frac{1-3\alpha}{\alpha+5}}$ |

$\beta \geq 1/2$ it is a *Hilbertian metric* and its square root satisfies the triangle inequality [14]. The second step is to bound $\mathbb{E}_{\mathbf{X}^N}\left[D_f(\hat{Q}_Z^N, Q_Z)\right]$ in terms of $\mathbb{E}_{\mathbf{X}^N}\left[\chi^2(\hat{Q}_Z^N, Q_Z)\right]$, which is the variance of the average of $N$ i.i.d. random variables and therefore decomposes as $\mathbb{E}_{X \sim Q_X}\left[\chi^2(Q_{Z|X}, Q_Z)\right]/N$. $\square$

**Theorem 2** (Rates of the bias). *If* $\mathbb{E}_{X \sim Q_X, Z \sim P_Z}\left[q^4(Z|X)/p^4(Z)\right]$ *is finite then the bias* $\mathbb{E}_{\mathbf{X}^N}\left[D_f(\hat{Q}_Z^N \| P_Z)\right] - D_f\left(Q_Z \| P_Z\right)$ *decays with rate as given in the second row of Table 1.*

*Proof sketch (full proof in Appendix B.4).* Denoting by $\hat{q}_N(z)$ the density of $\hat{Q}_Z^N$, the proof is based on the inequality $f\left(\hat{q}_N(z)/p(z)\right) - f\left(q(z)/p(z)\right) \leq \frac{\hat{q}_N(z)-q(z)}{p(z)} f'\left(\hat{q}_N(z)/p(z)\right)$ due to convexity of $f$, applied to the bias. The integral of this inequality is bounded by controlling $f'$, requiring subtle treatment when $f'$ diverges when the density ratio $\hat{q}_N(z)/p(z)$ approaches zero. $\square$

## 2.2 Tail bounds for RAM and practical estimation with RAM-MC

Theorems 1 and 2 describe the convergence of the *expectation* of RAM over $\mathbf{X}^N$, which in practice may be intractable. Fortunately, the following shows that RAM rapidly concentrates to its expectation.

**Theorem 3** (Tail bounds for RAM). *Suppose that* $\chi^2\left(Q_{Z|x} \| P_Z\right) \leq C < \infty$ *for all $x$ and for some constant $C$. Then, the RAM estimator $D_f(\hat{Q}_Z^N \| P_Z)$ concentrates to its mean in the following sense. For $N > 8$ and for any $\delta > 0$, with probability at least $1 - \delta$ it holds that*

$$\left| D_f(\hat{Q}_Z^N \| P_Z) - \mathbb{E}_{\mathbf{X}^N}\left[D_f(\hat{Q}_Z^N \| P_Z)\right] \right| \leq K \cdot \psi(N)\sqrt{\log(2/\delta)},$$

*where $K$ is a constant and $\psi(N)$ is given in Table 2.*

*Proof sketch (full proof in Appendix B.5).* These results follow by applying McDiarmid's inequality. To apply it we need to show that RAM viewed as a function of $\mathbf{X}^N$ has bounded differences. We show that when replacing $X_i \in \mathbf{X}^N$ with $X_i'$ the value of $D_f(\hat{Q}_Z^N \| P_Z)$ changes by at most $O(N^{-1/2}\psi(N))$. Proof of this proceeds similarly to the one of Theorem 2. $\square$

In practice it may not be possible to evaluate $D_f(\hat{Q}_Z^N \| P_Z)$ analytically. We propose to use Monte-Carlo (MC) estimation since both densities $\hat{q}_N(z)$ and $p(z)$ are assumed to be known. We consider importance sampling with proposal distribution $\pi(z|\mathbf{X}^N)$, highlighting the fact that $\pi$ can depend on the sample $\mathbf{X}^N$. If $\pi(z|\mathbf{X}^N) = p(z)$ this reduces to normal MC sampling. We arrive at the **RAM-MC estimator** based on $M$ i.i.d. samples $\mathbf{Z}^M := \{Z_1, \ldots, Z_M\}$ from $\pi(z|\mathbf{X}^N)$:

$$\hat{D}_f^M(\hat{Q}_Z^N \| P_Z) := \frac{1}{M}\sum_{m=1}^{M} f\left(\frac{\hat{q}_N(Z_m)}{p(Z_m)}\right)\frac{p(Z_m)}{\pi(Z_m|\mathbf{X}^N)}. \tag{3}$$

Table 3: Rate of bias for other estimators of $D_f(P, Q)$.

| $f$-divergence | KL | TV | $\chi^2$ | H$^2$ | JS | $D_{f_\beta}$ $\frac{1}{2}<\beta<1$ | $D_{f_\beta}$ $1<\beta<\infty$ | $D_{f_\alpha}$ $-1<\alpha<1$ |
|---|---|---|---|---|---|---|---|---|
| Krishnamurthy et al. [22] | - | - | - | - | - | - | - | $N^{-\frac{1}{2}}+N^{\frac{-3s}{2s+d}}$ |
| Nguyen et al. [28] | $N^{-\frac{1}{2}}$ | - | - | - | - | - | - | - |
| Moon and Hero [26] | $N^{-\frac{1}{2}}$ | - | $N^{-\frac{1}{2}}$ | $N^{-\frac{1}{2}}$ | $N^{-\frac{1}{2}}$ | $N^{-\frac{1}{2}}$ | $N^{-\frac{1}{2}}$ | $N^{-\frac{1}{2}}$ |

**Theorem 4** (RAM-MC is unbiased and consistent). $\mathbb{E}\big[\hat{D}_f^M(\hat{Q}_Z^N \| P_Z)\big] = \mathbb{E}\big[D_f(\hat{Q}_Z^N \| P_Z)\big]$ *for any proposal distribution* $\pi$. *If* $\pi(z|\mathbf{X}^N) = p(z)$ *or* $\pi(z|\mathbf{X}^N) = \hat{q}_N(z)$ *then under mild assumptions*$^\star$ *on the moments of* $q(Z|X)/p(Z)$ *and denoting by* $\psi(N)$ *the rate given in Table 2, we have*

$$Var_{\mathbf{X}^N, \mathbf{Z}^M}\big[\hat{D}_f^M(\hat{Q}_Z^N \| P_Z)\big] = O\left(M^{-1}\right) + O\left(\psi(N)^2\right).$$

*Proof sketch ($^\star$full statement and proof in Appendix B.6).* By the law of total variance,

$$\mathrm{Var}_{\mathbf{X}^N, \mathbf{Z}^M}\big[\hat{D}_f^M\big] = \mathbb{E}_{\mathbf{X}^N}\big[\mathrm{Var}\big[\hat{D}_f^M \,|\, \mathbf{X}^N\big]\big] + \mathrm{Var}_{\mathbf{X}^N}\big[D_f(\hat{Q}_Z^N \| P_Z)\big].$$

The first of these terms is $O(M^{-1})$ by standard results on MC integration, subject to the assumptions on the moments. Using the fact that $\mathrm{Var}[Y] = \int_0^\infty \mathbb{P}(|Y - \mathbb{E}Y| > \sqrt{t})dt$ for any random variable $Y$ we bound the second term by integrating the exponential tail bound of Theorem 3. $\square$

Through use of the Efron-Stein inequality—rather than integrating the tail bound provided by McDiarmid's inequality—it is possible for some choices of $f$ to weaken the assumptions under which the $O(\psi(N)^2)$ variance is achieved: from uniform boundedness of $\chi^2(Q_{Z|X} \| P_Z)$ to boundedness in expectation. In general, a variance better than $O(M^{-1})$ is not possible using importance sampling. However, the constant and hence practical performance may vary significantly depending on the choice of $\pi$. We note in passing that through Chebyshev's inequality, it is possible to derive confidence bounds for RAM-MC of the form similar to Theorem 3, but with an additional dependence on $M$ and worse dependence on $\delta$. For brevity we omit this.

### 2.3 Discussion: assumptions and summary

All the rates in this section are independent of the dimension of the space $\mathcal{Z}$ over which the distributions are defined. However the constants may exhibit some dependence on the dimension. Accordingly, for fixed $N$, the bias and variance may generally grow with the dimension.

Although the data distribution $Q_X$ will generally be unknown, in some practical scenarios such as deep autoencoder models, $P_Z$ may be chosen by design and $Q_{Z|X}$ learned subject to architectural constraints. In such cases, the assumptions of Theorems 2 and 3 can be satisfied by making suitable restrictions (we conjecture also for Theorem 1). For example, suppose that $P_Z$ is $\mathcal{N}(0, I_d)$ and $Q_{Z|X}$ is $\mathcal{N}(\mu(X), \Sigma(X))$ with $\Sigma$ diagonal. Then the assumptions hold if there exist constants $K, \epsilon > 0$ such that $\|\mu(X)\| < K$ and $\Sigma_{ii}(X) \in [\epsilon, 1]$ for all $i$ (see Appendix B.7). In practice, numerical stability often requires the diagonal entries of $\Sigma$ to be lower bounded by a small number (e.g. $10^{-6}$). If $\mathcal{X}$ is compact (as for images) then such a $K$ is guaranteed to exist; if not, choosing $K$ very large yields an insignificant constraint.

Table 3 summarizes the rates of bias for some existing methods. In contrast to our proposal, the assumptions of these estimators may in practice be difficult to verify. For the estimator of [22], both densities $p$ and $q$ must belong to the Hölder class of smoothness $s$, be supported on $[0, 1]^d$ and satisfy $0 < \eta_1 < p, q < \eta_2 < \infty$ on the support for known constants $\eta_1, \eta_2$. For that of [28], the density ratio $p/q$ must satisfy $0 < \eta_1 < p/q < \eta_2 < \infty$ and belong to a function class $G$ whose *bracketing entropy* (a measure of the complexity of a function class) is properly bounded. The condition on the bracketing entropy is quite strong and ensures that the density ratio is well behaved. For the estimator of [26], both $p$ and $q$ must have the same bounded support and satisfy $0 < \eta_1 < p, q < \eta_2 < \infty$ on the support. $p$ and $q$ must have *continuous bounded* derivatives of order $d$ (which is stronger than assumptions of [22]), and $f$ must have derivatives of order at least $d$.

In summary, the RAM estimator $D_f(\hat{Q}_Z^N \| P_Z)$ for $D_f(Q_Z \| P_Z)$ is **consistent** since it concentrates to its expectation $\mathbb{E}_{\mathbf{X}^N}[D_f(\hat{Q}_Z^N \| P_Z)]$, which in turn converges to $D_f(Q_Z \| P_Z)$. It is also **practical** because it can be efficiently estimated with Monte-Carlo sampling via RAM-MC.

## 3 Empirical evaluation

In the previous section we showed that our proposed estimator has a number of desirable theoretical properties. Next we demonstrate its practical performance. First, we present a synthetic experiment investigating the behaviour of RAM-MC in controlled settings where all distributions and divergences are known. Second, we investigate the use of RAM-MC in a more realistic setting to estimate a divergence between the aggregate posterior $Q_Z$ and prior $P_Z$ in pretrained autoencoder models. For experimental details not included in the main text, see Appendix C[2].

### 3.1 Synthetic experiments

**The data model.** Our goal in this subsection is to test the behaviour of the RAM-MC estimator for various $d = \dim(\mathcal{Z})$ and $f$-divergences. We choose a setting in which $Q_Z^\lambda$ parametrized by a scalar $\lambda$ and $P_Z$ are both $d$-variate normal distributions for $d \in \{1, 4, 16\}$. We use RAM-MC to estimate $D_f(Q_Z^\lambda, P_Z)$, which can be computed analytically for the KL, $\chi^2$, and squared Hellinger divergences in this setting (see Appendix C.1.1). Namely, we take $P_Z$ and $Q_X$ to be standard normal distributions over $\mathcal{Z} = \mathbb{R}^d$ and $\mathcal{X} = \mathbb{R}^{20}$ respectively, and $Z \sim Q_{Z|X}^\lambda$ be a linear transform of $X$ plus a fixed isotropic Gaussian noise, with the linear function parameterized by $\lambda$. By varying $\lambda$ we can interpolate between different values for $D_f(Q_Z^\lambda \| P_Z)$.

**The estimators.** In Figure 1 we show the behaviour of RAM-MC with $N \in \{1, 500\}$ and $M = 128$ compared to the ground truth as $\lambda$ is varied. The columns of Figure 1 correspond to different dimensions $d \in \{1, 4, 16\}$, and rows to the KL, $\chi^2$ and H$^2$ divergences, respectively. We also include two baseline methods. First, a plug-in method based on kernel density estimation [26]. Second, and only for the KL case, the M1 method of [28] based on density ratio estimation.

**The experiment.** To produce each plot, the following was performed 10 times, with the mean result giving the bold lines and standard deviation giving the error bars. First, $N$ points $\mathbf{X}^N$ were drawn from $Q_X$. Then $M = 128$ points $\mathbf{Z}^M$ were drawn from $\hat{Q}_Z^N$ and RAM-MC (3) was evaluated. For the plug-in estimator, the densities $\hat{q}(z)$ and $\hat{p}(z)$ were estimated by kernel density estimation with 500 samples from $Q_Z$ and $P_Z$ respectively using the default settings of the Python library `scipy.stats.gaussian_kde`. The divergence was then estimated via MC-sampling using 128 samples from $Q_Z$ and the surrogate densities. The M1 estimator involves solving a convex linear program in $N$ variables to maximize a lower bound on the true divergence, see [28] for more details. Although the M1 estimator can in principle be used for arbitrary $f$-divergences, its implementation requires hand-crafted derivations that are supplied only for the KL in [28], which are the ones we use.

**Discussion.** The results of this experiment empirically support Proposition 1 and Theorems 1, 2, and 4: (i) in expectation, RAM-MC upper bounds the true divergence; (ii) by increasing $N$ from 1 to 500 we clearly decrease both the bias and the variance of RAM-MC. When the dimension $d$ increases, the bias for fixed $N$ also increases. This is consistent with the theory in that, although the rates are independent of $d$, the constants are not. We note that by side-stepping the issue of density estimation, RAM-MC performs favourably compared to the plug-in and M1 estimators, more so in higher dimensions ($d = 16$). In particular, the shape of the RAM-MC curve follows that of the truth for each divergence, while that of the plug-in estimator does not for larger dimensions. In some cases the plug-in estimator can even take negative values because of the large variance.

### 3.2 Real-data experiments

**The data model.** To investigate the behaviour of RAM-MC in a more realistic setting, we consider Variational Autoencoders (VAEs) and Wasserstein Autoencoders (WAEs) [21, 38]. Both models involve learning an *encoder* $Q_{Z|X}^\theta$ with parameter $\theta$ mapping from high dimensional data to a lower dimensional latent space and decoder mapping in the reverse direction. A prior distribution

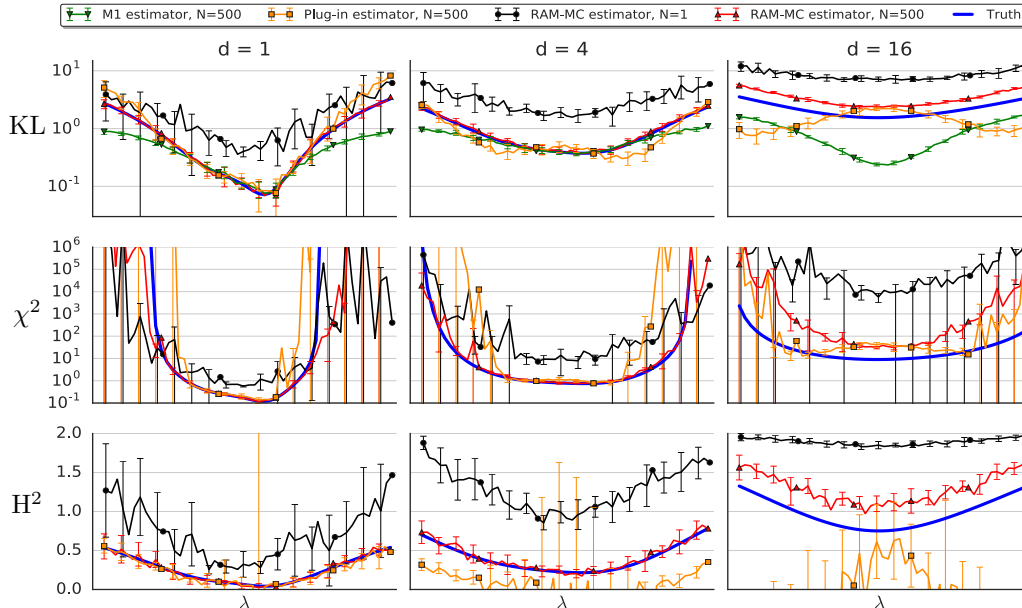

Figure 1: (Section 3.1) Estimating $D_f\big(\mathcal{N}(\mu_\lambda, \Sigma_\lambda), \mathcal{N}(0, I_d)\big)$ for various $f$, $d$, and parameters $\mu_\lambda$ and $\Sigma_\lambda$ indexed by $\lambda \in \mathbb{R}$. Horizontal axis correspond to $\lambda \in [-2, 2]$, columns to $d \in \{1, 4, 16\}$ and rows to KL, $\chi^2$, and H$^2$ divergences respectively. **Blue** are true divergences, **black** and **red** are RAM-MC estimators (3) for $N \in \{1, 500\}$ respectively, **green** are M1 estimator of [28] and **orange** are plug-in estimates based on Gaussian kernel density estimation [26]. $N = 500$ and $M = 128$ in all the plots if not specified otherwise. Error bars depict one standard deviation over 10 experiments.

$P_Z$ is specified, and the optimization objectives of both models are of the form "reconstruction + distribution matching penalty". The penalty of the VAE was shown by [19] to be equivalent to $\text{KL}(Q_Z^\theta \| P_Z) + I(X, Z)$ where $I(X, Z)$ is the mutual information of a sample and its encoding. The WAE penalty is $D(Q_Z^\theta \| P_Z)$ for any divergence $D$ that can practically be estimated. Following [38], we trained models using the Maximum Mean Discrepency (MMD), a kernel-based distance on distributions, and a divergence estimated using a GAN-style classifier leading to WAE-MMD and WAE-GAN respectively [13, 12]. For more information about VAE and WAE, see Appendix C.2.1.

**The experiment.** We consider models pre-trained on the *CelebA* dataset [25], and use them to evaluate the RAM-MC estimator as follows. We take the test dataset as the ground-truth $Q_X$, and embed it into the latent space via the trained encoder. As a result, we obtain a $\sim$20k-component Gaussian mixture for $Q_Z$, the *empirical aggregate posterior*. Since $Q_Z$ is a finite—not continuous— mixture, the true $D_f(Q_Z \| P_Z)$ can be estimated using a large number of MC samples (we used $10^4$). Note that this is very costly and involves evaluating $2 \cdot 10^4$ Gaussian densities for each of the $10^4$ MC points. We repeated this evaluation 10 times and report means and standard deviations. RAM-MC is evaluated using $N \in \{2^0, 2^1, \ldots, 2^{14}\}$ and $M \in \{10, 10^3\}$. For each combination $(N, M)$, RAM-MC was computed 50 times with the means plotted as bold lines and standard deviations as error bars. In Figure 2 we show the result of performing this for the KL divergence on six different models. For each dimension $d \in \{32, 64, 128\}$, we chose two models from the classes (VAE, WAE-MMD, WAE-GAN). See Appendix C.2 for further details and similar plots for the $H^2$-divergence.

**Discussion.** The results are encouraging. In all cases RAM-MC achieves a reasonable accuracy with $N$ relatively small, even for the bottom right model where the true KL divergence ($\approx 1910$) is very big. We see evidence supporting Theorem 4, which says that the variance of RAM-MC is mostly determined by the smaller of $\psi(N)$ and $M$: when $N$ is small, the variance of RAM-MC does not change significantly with $M$, however when $N$ is large, increasing $M$ significantly reduces the variance. Also we found there to be two general modes of behaviour of RAM-MC across the six trained models we considered. In the bottom row of Figure 2 we see that the decrease in bias with

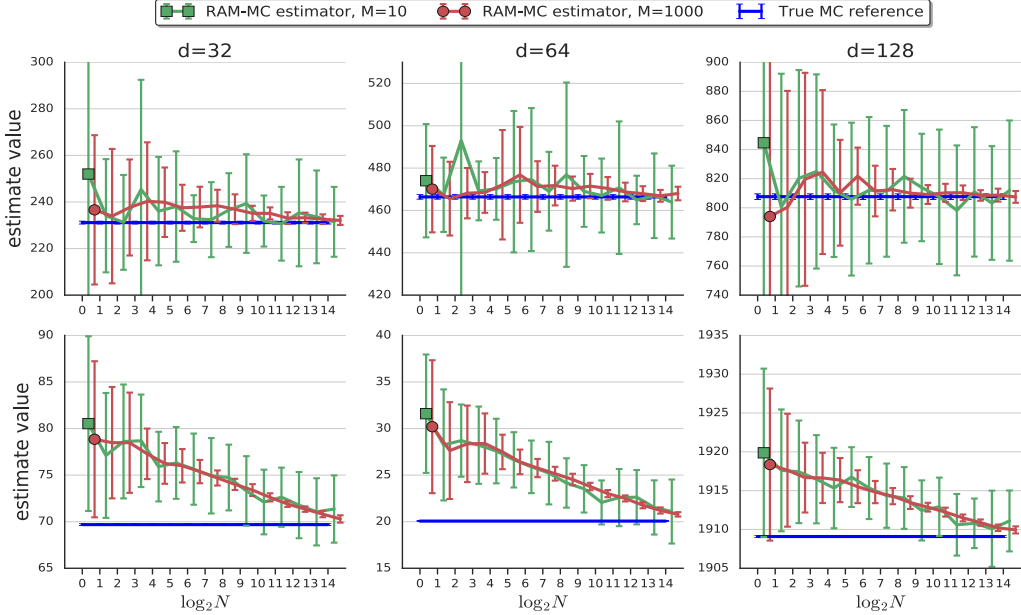

Figure 2: (Section 3.2) Estimates of $KL(Q_Z^\theta \| P_Z)$ for pretrained autoencoder models with RAM-MC as a function of $N$ for $M{=}10$ (**green**) and $M{=}1000$ (**red**) compared to an accurate MC estimate of the ground truth (**blue**). Lines and error bars represent means and standard deviations over 50 trials.

$N$ is very obvious, supporting Proposition 1 and Theorems 1 and 2. In contrast, in the top row it is less obvious, because the comparatively larger variance for $M{=}10$ dominates reductions in the bias. Even in this case, both the bias and variance of RAM-MC with $M{=}1000$ become negligible for large $N$. Importantly, the behaviour of RAM-MC does not degrade in higher dimensions.

The baseline estimators (plug-in [26] and M1 [28]) perform so poorly that we decided not to include them in the plots (doing so would distort the $y$-axis scale). In contrast, even with a relatively modest $N{=}2^8$ and $M{=}1000$ samples, RAM-MC behaves reasonably well in all cases.

## 4    Applications: total correlation, entropy, and mutual information estimates

In this section we describe in detail some direct consequences of our new estimator and its guarantees. Our theory may also apply to a number of machine learning domains where estimating entropy, total correlation or mutual information is either the final goal or part of a broader optimization loop.

**Total correlation and entropy estimation.**    The differential entropy, which is defined as $H(Q_Z) = -\int_{\mathcal{Z}} q(z) \log q(z) dz$, is often a quantity of interest in machine learning. While this is intractable in general, straightforward computation shows that for *any* $P_Z$

$$H(Q_Z) - \mathbb{E}_{\mathbf{X}^N} H(\hat{Q}_Z^N) = \mathbb{E}_{\mathbf{X}^N} \mathrm{KL}[\hat{Q}_Z^N \| P_Z] - \mathrm{KL}[Q_Z \| P_Z].$$

Therefore, our results provide sufficient conditions under which $H(\hat{Q}_Z^N)$ converges to $H(Q_Z)$ and concentrates to its mean. We now examine some consequences for Variational Autoencoders (VAEs).

Total Correlation is considered by [6], $TC(Q_Z) := \mathrm{KL}[Q_Z \| \prod_{i=1}^{d_Z} Q_{Z_i}] = \sum_{i=1}^{d_Z} H(Q_{Z_i}) - H(Q_Z)$ where $Q_{Z_i}$ is the $i$th marginal of $Q_Z$. This is added to the VAE loss function to encourage $Q_Z$ to be factorized, resulting in the $\beta$-TC-VAE algorithm. By the second equality above, estimation of TC can be reduced to estimation of $H(Q_Z)$ (only slight modifications are needed to treat $H(Q_{Z_i})$).

Two methods are proposed in [6] for estimating $H(Q_Z)$, both of which assume a finite dataset of size $D$. One of these, named *Minibatch Weighted Sample* (MWS), coincides with $H(\hat{Q}_Z^N) + \log D$ estimated with a particular form of MC sampling. Our results therefore imply *inconsistency* of the MWS method due to the constant $\log D$ offset. In the context of [6] this is not actually problematic

since a constant offset does not affect gradient-based optimization techniques. Interestingly, although the derivations of [6] suppose a data distribution of finite support, our results show that minor modifications result in an estimator suitable for both finite and infinite support data distributions.

**Mutual information estimation.** The mutual information (MI) between variables with joint distribution $Q_{Z,X}$ is defined as $I(Z,X) := \mathrm{KL}\left[Q_{Z,X}\|Q_Z Q_X\right] = \mathbb{E}_X \mathrm{KL}\left[Q_{Z|X}\|Q_Z\right]$. Several recent papers have estimated or optimized this quantity in the context of autoencoder architectures, coinciding with our setting [8, 19, 1, 31]. In particular, [36] propose the following estimator based on replacing $Q_Z$ with $\hat{Q}_Z^N$, proving it to be a lower bound on the true MI:

$$I_{TCPC}^N(Z,X) = \mathbb{E}_{\mathbf{X}^N}\left[\frac{1}{N}\sum_{i=1}^N \mathrm{KL}[Q_{Z|X_i}\|\hat{Q}_Z^N]\right] \le I(Z,X).$$

The gap can be written as $I(Z,X) - I_{TCPC}^N(Z,X) = \mathbb{E}_{\mathbf{X}^N} \mathrm{KL}[\hat{Q}_Z^N\|P_Z] - \mathrm{KL}[Q_Z\|P_Z]$ where $P_Z$ is *any* distribution. Therefore, our results also provide sufficient conditions under which $I_{TCPC}^N$ converges and concentrates to the true mutual information.

# 5 Conclusion

We introduced a practical estimator for the $f$-divergence $D_f(Q_Z\|P_Z)$ where $Q_Z = \int Q_{Z|X} dQ_X$, samples from $Q_X$ are available, and $P_Z$ and $Q_{Z|X}$ have known density. The RAM estimator is based on approximating the true $Q_Z$ with data samples as a random mixture via $\hat{Q}_Z^N = \frac{1}{N}\sum_n Q_{Z|X_n}$. We denote by RAM-MC the estimator version where $D_f(\hat{Q}_Z^N\|P_Z)$ is estimated with MC sampling. We proved rates of convergence and concentration for both RAM and RAM-MC, in terms of sample size $N$ and MC samples $M$ under a variety of choices of $f$. Synthetic and real-data experiments strongly support the validity of our proposal in practice, and our theoretical results provide guarantees for methods previously proposed heuristically in existing literature.

Future work will investigate the use of our proposals for optimization loops, in contrast to pure estimation. When $Q_{Z|X}^\theta$ depends on parameter $\theta$ and the goal is to minimize $D_f(Q_Z^\theta\|P_Z)$ with respect to $\theta$, RAM-MC provides a practical surrogate loss that can be minimized using stochastic gradient methods.

### Acknowledgements

Thanks to Alessandro Ialongo, Niki Kilbertus, Luigi Gresele, Giambattista Parascandolo, Mateo Rojas-Carulla and the rest of Empirical Inference group at the MPI, and Ben Poole, Sylvain Gelly, Alexander Kolesnikov and the rest of the Brain Team in Zurich for stimulating discussions, support and advice.

## Footnotes

[2] A python notebook to reproduce all experiments is available at https://github.com/google-research/google-research/tree/master/f_divergence_estimation_ram_mc.

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
