[Supplementary Material]

Table 4: $f$ corresponding to divergences referenced in this paper.

| $f$-divergence | $f_0(x)$ |
|---|---|
| KL | $x \log x - x + 1$ |
| TV | $\frac{1}{2}\|1 - x\|$ |
| $\chi^2$ | $x^2 - 2x$ |
| H$^2$ | $2(1 - \sqrt{x})$ |
| JS | $(1 + x) \log(\frac{2}{1+x}) + x \log x$ |
| $D_{f_\beta}, \beta > 0, \beta \neq \frac{1}{2}$ | $\frac{1}{1-\frac{1}{\beta}} \left[ (1 + x^\beta)^{\frac{1}{\beta}} - 2^{\frac{1}{\beta}-1}(1 + x) \right]$ |
| $D_{f_\alpha}, -1 < \alpha < 1$ | $\frac{4}{1-\alpha^2} \left( 1 - x^{\frac{1+\alpha}{2}} \right) - \frac{2(x-1)}{\alpha-1}$ |

## A $f$ for divergences considered in this paper

One of the useful properties of $f$-divergences that we make use of in the proofs of Theorems 2 and 3 is that for any constant $c$, replacing $f(x)$ by $f(x) + c(x - 1)$ does not change the divergence $D_f$. It is often convenient to work with $f_0(x) := f(x) - f'(1)(x - 1)$ which is decreasing on $(0, 1)$ and increasing on $(1, \infty)$ and satisfies $f'_0(1) = 0$.

In Table 4 we list the forms of the function $f_0$ for each of the divergences considered in this paper.

## B Proofs

### B.1 Proof of Proposition 1

**Proposition 1.** *Let $M \leq N$ be integers. Then*

$$D_f(Q_Z \| P_Z) \leq \mathbb{E}_{\mathbf{X}^N \sim Q_X^N} D_f(\hat{Q}_Z^N \| P_Z) \leq \mathbb{E}_{\mathbf{X}^M \sim Q_X^M} D_f(\hat{Q}_Z^M \| P_Z).$$

*Proof.* Observe that $\mathbb{E}_{\mathbf{X}^N} \hat{Q}_Z^N = Q_Z$. Thus,

$$
\begin{aligned}
D_f(Q_Z \| P_Z) &= \int f \left( \frac{\mathbb{E}_{\mathbf{X}^N} \hat{q}_N(z)}{p(z)} \right) dP_Z(z) \\
&\leq \mathbb{E}_{\mathbf{X}^N} \int f \left( \frac{\hat{q}_N(z)}{p(z)} \right) dP_Z(z) \\
&= \mathbb{E}_{\mathbf{X}^N \sim P_X^N} D_f(\hat{Q}_Z^N \| P_Z),
\end{aligned}
$$

where the inequality follows from convexity of $f$.

To see that $\mathbb{E}_{\mathbf{X}^N \sim P_X^N} D_f(\hat{Q}_Z^N \| P_Z) \leq \mathbb{E}_{\mathbf{X}^M \sim P_X^N} D_f(\hat{Q}_Z^M \| P_Z)$ for $N \geq M$, let $I \subseteq \{1, \dots, N\}, |I| = M$ and write

$$\hat{Q}_Z^I = \frac{1}{M} \sum_{i \in I} Q_{Z|X_i}.$$

Letting $I$ be a random subset chosen uniformly *without replacement*, observe that for any fixed $I$, $\mathbf{X}^I \sim \mathbb{P}_X^M$ (with the randomness coming from $\mathbf{X}^N \sim \mathbb{P}_X^N$). Thus

$$\hat{Q}_Z^N = \frac{1}{N} \sum_{i=1}^{N} Q_{Z|X_i}$$

$$= \mathbb{E}_I \frac{1}{M} \sum_{i \in I} Q_{Z|X_i}$$

$$= \mathbb{E}_I \hat{Q}_Z^I$$

and so again by convexity of $f$ we have that

$$\mathbb{E}_{\mathbf{X}^N \sim P_X^N} D_f(\hat{Q}_Z^N \| P_Z) \leq \mathbb{E}_{\mathbf{X}^N} \mathbb{E}_I D_f(\hat{Q}_Z^I \| P_Z) \tag{4}$$

$$= \mathbb{E}_{\mathbf{X}^M} D_f(\hat{Q}_Z^M \| P_Z) \tag{5}$$

with the last line following from the observation that $\mathbf{X}^I \sim \mathbb{P}_X^M$. □

## B.2   Proof of Theorem 1

**Lemma 1.** *Suppose that $D_f^{\frac{1}{2}}$ satisfies the triangle inequality. Then for any $\lambda > 0$,*

$$D_f\left(\hat{Q}_Z^N \| P_Z\right) - D_f\left(Q_Z \| P_Z\right) \leq (1+\lambda) D_f\left(\hat{Q}_Z^N \| Q_Z\right) + \frac{1}{\lambda} D_f\left(Q_Z \| P_Z\right)$$

*If, furthermore, $\mathbb{E}_{\mathbf{X}^N}\left[D_f\left(\hat{Q}_Z^N \| Q_Z\right)\right] = O\left(\frac{1}{N^k}\right)$ for some $k > 0$, then*

$$\mathbb{E}_{\mathbf{X}^N}\left[D_f\left(\hat{Q}_Z^N \| P_Z\right)\right] - D_f\left(Q_Z \| P_Z\right) = O\left(\frac{1}{N^{k/2}}\right)$$

*Proof.* The first inequality follows from the triangle inequality for $D_f^{\frac{1}{2}}$ on $\hat{Q}_Z^N$ and $P_Z$, and the fact that $2\sqrt{ab} \leq \lambda a + \frac{b}{\lambda}$ for $a, b, \lambda > 0$. The second inequality follows from the first by taking $\lambda = N^{-\frac{k}{2}}$. □

**Theorem 1** (Rates of the bias). *If $\mathbb{E}_{X \sim Q_X}\left[\chi^2\left(Q_{Z|X}, Q_Z\right)\right]$ and $\mathrm{KL}\left(Q_Z \| P_Z\right)$ are finite then the bias $\mathbb{E}_{\mathbf{X}^N}\left[D_f(\hat{Q}_Z^N \| P_Z)\right] - D_f\left(Q_Z \| P_Z\right)$ decays with rate as given in the first row of Table 1.*

*Proof.* To begin, observe that

$$\mathbb{E}_{\mathbf{X}^N}\left[\chi^2(\hat{Q}_Z^N, Q_Z)\right] = \mathbb{E}_{\mathbf{X}^N} \mathbb{E}_{Q_Z}\left[\left(\frac{\hat{q}_N(z)}{q(z)} - 1\right)^2\right]$$

$$= \mathbb{E}_{Q_Z} \mathrm{Var}_{\mathbf{X}^N}\left[\frac{1}{N} \sum_{n=1}^{N} \frac{q(z|X_n)}{q(z)}\right]$$

$$= \frac{1}{N} \mathbb{E}_{Q_Z} \mathrm{Var}_X\left[\frac{q(z|X)}{q(z)}\right]$$

$$= \frac{1}{N} \mathbb{E}_X\left[\chi^2(Q_{Z|X}, Q_Z)\right]$$

where the introduction of the variance operator follows from the fact that $\mathbb{E}_{X_N}\left[\frac{\hat{q}_N(z)}{q(z)}\right] = 1$.

For the KL-divergence, using the fact that $\mathrm{KL} \leq \chi^2$ (Lemma 2.7 of [2]) yields

$$\mathbb{E}_{\mathbf{X}^N}\left[\mathrm{KL}\left(\hat{Q}_Z^N \| P_Z\right)\right] - \mathrm{KL}\left(Q_Z \| P_Z\right) = \mathbb{E}_{\mathbf{X}^N}\left[\mathrm{KL}\left(\hat{Q}_Z^N \| Q_Z\right)\right]$$

$$\leq \mathbb{E}_{\mathbf{X}^N}\left[\chi^2(\hat{Q}_Z^N, Q_Z)\right]$$

$$= \frac{1}{N} \mathbb{E}_X\left[\chi^2(Q_{Z|X}, Q_Z)\right]$$

$$= O\left(\frac{1}{N}\right),$$

where the first equality can be verified by using the definition of KL and the fact that $Q_Z = \mathbb{E}_{\mathbf{X}^N} \hat{Q}_Z^N$.

For Total Variation, we have

$$\mathbb{E}_{\mathbf{X}^N}\left[\mathrm{TV}\left(\hat{Q}_Z^N\|P_Z\right)\right] - \mathrm{TV}\left(Q_Z\|P_Z\right) \le \mathbb{E}_{\mathbf{X}^N}\left[\mathrm{TV}\left(\hat{Q}_Z^N\|Q_Z\right)\right]$$

$$\le \frac{1}{\sqrt{2}}\sqrt{\mathbb{E}_{\mathbf{X}^N}\left[\mathrm{KL}\left(\hat{Q}_Z^N\|Q_Z\right)\right]}$$

$$= O\left(\frac{1}{\sqrt{N}}\right),$$

where the first inequality holds since TV is a metric and thus obeys the triangle inequality, and the second inequality follows by Pinsker's inequality combined with concavity of $\sqrt{x}$ (Lemma 2.5 of [2]).

For $D_{f_\beta}$ (including Jenson-Shannon) using the fact that $D_{f_\beta}^{1/2}$ satisfies the triangular inequality, we apply the second part of Lemma 1 in combination with the fact that $D_{f_\beta}\left(\hat{Q}_Z^N\|Q_Z\right) \le \psi(\beta)\,\mathrm{TV}\left(\hat{Q}_Z^N\|Q_Z\right)$ for some scalar $\psi(\beta)$ (Theorem 2 of [32]) to obtain

$$\mathbb{E}_{\mathbf{X}^N}\left[D_{f_\beta}\left(\hat{Q}_Z^N\|P_Z\right)\right] - D_{f_\beta}\left(Q_Z\|P_Z\right) \le O\left(\frac{1}{N^{1/4}}\right).$$

Although the squared Hellinger divergence is a member of the $f_\beta$-divergence family, we can use the tighter bound $\mathrm{H}^2\left(\hat{Q}_Z^N\|Q_Z\right) \le KL\left(\hat{Q}_Z^N\|Q_Z\right)$ (Lemma 2.4 of [2]) in combination with Lemma 1 to obtain

$$\mathbb{E}_{\mathbf{X}^N}\left[\mathrm{H}^2\left(\hat{Q}_Z^N\|P_Z\right)\right] - \mathrm{H}^2\left(Q_Z\|P_Z\right) \le O\left(\frac{1}{\sqrt{N}}\right).$$

$\square$

## B.3 Upper bounds of f

We will make use of the following lemmas in the proof of Theorem 2 and 3.

**Lemma 2.** *Let* $f_0(x) = x\log x - x + 1$, *corresponding to* $D_{f_0} = \mathrm{KL}$. *Write* $g(x) = f_0'^2(x) = \log^2(x)$.
*For any* $0 < \delta < 1$, *the function*

$$h_\delta(x) := \begin{cases} g(\delta) + xg'(e) & x \in [0, e] \\ g(\delta) + eg'(e) + g(x) - g(e) & x \in [e, \infty) \end{cases}$$

*is an upper bound of* $g(x)$ *on* $[\delta, \infty)$, *and is concave and non-negative on* $[0, \infty)$.

*Proof.* First observe that $h_\delta$ is concave. It has continuous first and second derivatives:

$$h_\delta'(x) = \begin{cases} g'(e) & x \in [0, e] \\ g'(x) & x \in [e, \infty) \end{cases} \qquad h_\delta''(x) = \begin{cases} 0 & x \in [0, e] \\ g''(x) & x \in [e, \infty) \end{cases}$$

Note that $g''(x) = \frac{2}{x^2} - \frac{2\log(x)}{x^2} \le 0$ for $x \ge e$ and $g''(e) = 0$. Therefore $h_\delta''(x)$ has non-positive second derivative on $[0, \infty)$ and is thus concave on this set.

To see that $h_\delta(x)$ is an upper bound of $g(x)$ for $x \in [\delta, \infty)$, use the fact that $g'(x) = \frac{2\log(x)}{x}$ and observe that

$$h_\delta(x) - g(x) = \begin{cases} \log^2(\delta) + \frac{2x}{e} - \log^2(x) & x \in [\delta, e] \\ \log^2(\delta) + 1 & x \in [e, \infty) \end{cases} > 0.$$

To see that $h_\delta(x)$ is non-negative on $[0, \infty)$, note that $h_\delta(x) > g(x) \ge 0$ on $[\delta, \infty)$. Moreover, $g'(e) = 2/e > 0$, and so for $x \in [0, \delta]$ we have that $h_\delta(x) = g(\delta) + 2x/e \ge g(\delta) \ge 0$. $\square$

**Lemma 3.** *Let* $f_0(x) = 2(1 - \sqrt{x})$ *corresponding to the square of the Hellinger distance. Write* $g(x) = f_0'^2(x) = (1 - \frac{1}{\sqrt{x}})^2$. *For any* $0 < \delta < 1$, *the function*

$$h_\delta(x) = \frac{1}{\delta}(x - 1)^2$$

*is an upper bound of* $g(x)$ *on* $[\delta, \infty)$.

*Proof.* For $x = 1$, we have $g(1) = h_\delta(1)$. For $x \neq 1$,

$$0 \leq \frac{1}{\delta}(x-1)^2 - \left(1 - \frac{1}{\sqrt{x}}\right)^2$$

$$\iff \sqrt{\delta} \leq \frac{x-1}{1 - \frac{1}{\sqrt{x}}}$$

If $x \in [\delta, 1)$ then

$$\frac{x-1}{1 - \frac{1}{\sqrt{x}}} = \sqrt{x} \cdot \frac{\frac{1}{\sqrt{x}} - \sqrt{x}}{\frac{1}{\sqrt{x}} - 1} \geq \sqrt{x} \geq \sqrt{\delta}.$$

If $x \in (1, \infty)$ then

$$\frac{x-1}{1 - \frac{1}{\sqrt{x}}} = \sqrt{x} \cdot \frac{\sqrt{x} - \frac{1}{\sqrt{x}}}{1 - \frac{1}{\sqrt{x}}} \geq \sqrt{x} \geq \sqrt{\delta}.$$

Thus $g(x) \leq h_\delta(x)$ for $x \in [\delta, \infty)$. $\qquad\square$

**Lemma 4.** *Let* $f_0(x) = \frac{4}{1-\alpha^2}\left(1 - x^{\frac{1+\alpha}{2}}\right) - \frac{2(x-1)}{\alpha-1}$ *corresponding to the $\alpha$-divergence with $\alpha \in (-1, 1)$.*
*Write* $g(x) = f_0'^2(x) = \frac{4}{(\alpha-1)^2}\left(x^{\frac{\alpha-1}{2}} - 1\right)^2$*. For any $0 < \delta < 1$, the function*

$$h_\delta(x) = \frac{4\left(\delta^{\frac{\alpha-1}{2}} - 1\right)^2}{(\alpha-1)^2(\delta-1)^2} \cdot (x-1)^2$$

*is an upper bound of $g(x)$ on $[\delta, \infty)$.*

*Proof.* For $x = 1$, we have $g(1) = h_\delta(1)$. Consider now the case that $x \geq \delta$ and $x \neq 1$. Since $0 < \delta < 1$, we have that $1 - \delta > 0$. And because $(\alpha - 1)/2 \in (-1, 0)$, we have that $\delta^{\frac{\alpha-1}{2}} - 1 > 0$. It follows by taking square roots that

$$g(x) \leq h_\delta(x)$$

$$\iff d(x) := \frac{x^{\frac{\alpha-1}{2}} - 1}{1 - x} \leq \frac{\delta^{\frac{\alpha-1}{2}} - 1}{1 - \delta}$$

Now, $d(x)$ is non-increasing for $x > 0$. Indeed,

$$d'(x) = \frac{-1}{(1-x)^2}\left[1 - \frac{3-\alpha}{2}x^{\frac{\alpha-1}{2}} + \frac{1-\alpha}{2}x^{\frac{\alpha-3}{2}}\right]$$

and it can be shown by differentiating that the term inside the square brackets attains its minimum at $x = 1$ and is therefore non-negative. Since $(1-x)^2 \geq 0$ it follows that $d'(x) \leq 0$ and so $d(x)$ is non-increasing. From this fact it follows that $d(x)$ attains its maximum on $x \in [\delta, \infty)$ at $x = \delta$, and thus the desired inequality holds. $\qquad\square$

**Lemma 5.** *Let* $f_0(x) = (1 + x)\log 2 + x\log x - (1 + x)\log(1 + x)$ *corresponding to the Jensen-Shannon divergence. Write* $g(x) = f_0'^2(x) = \log^2 2 + \log^2\left(\frac{x}{1+x}\right) + 2\log 2\log\left(\frac{x}{1+x}\right)$*. For $0 < \delta < 1$, the function*

$$h_\delta(x) = g(\delta) + 4\log^2 2$$

*is an upper bound of $g(x)$ on $[\delta, \infty)$.*

*Proof.* For $x \geq 1$, $\frac{x}{x+1} \in [0.5, 1)$ and so $\log\left(\frac{x}{1+x}\right) \in [-\log 2, 0)$. Therefore $g(x) \in \left(0, 4\log^2 2\right]$ for $x > 1$. It follows that for any value of $\delta$, $h_\delta(x) \geq g(x)$ for $x \geq 1$. $f_0'(1) = 0$ and by differentiating again it can be shown that $f_0''(x) > 0$ for $x \in (0, 1)$. Thus $f_0'(x) < 0$ and is increasing on $(0, 1)$ and so $g(x) > 0$ and is decreasing on $(0, 1)$. Thus $h_\delta(x) > g(\delta) \geq g(x)$ for $x \in [\delta, 1)$. $\qquad\square$

**Lemma 6.** *Let* $f_0(x) = \frac{1}{1-\frac{1}{\beta}}\left[(1 + x^\beta)^{\frac{1}{\beta}} - 2^{\frac{1}{\beta}-1}(1 + x)\right]$ *corresponding to the $f_\beta$-divergence introduced in [32]. We assume $\beta \in \left(\frac{1}{2}, \infty\right) \setminus \{1\}$. Write* $g(x) = f_0'^2(x) = \left(\frac{\beta}{1-\beta}\right)^2\left[(1 + x^{-\beta})^{\frac{1-\beta}{\beta}} - 2^{\frac{1}{\beta}-1}\right]^2$*.*

*If $\beta \in \left(\frac{1}{2}, 1\right)$, then $\lim_{x\to\infty} g(x)$ exists and is finite and for any $0 < \delta < 1$, we have that $h_\delta(x) := g(\delta) + \lim_{x\to\infty} g(x) \geq g(x)$ for all $x \in [\delta, \infty)$.*

*If $\beta \in (1, \infty)$, then $\lim_{x\to 0} g(x)$ and $\lim_{x\to\infty} g(x)$ both exist and are finite, and $g(x) \leq \max\{\lim_{x\to 0} g(x), \lim_{x\to\infty} g(x)\}$ for all $x \in [0, \infty)$.*

*Proof.* For any $\beta \in \left(\frac{1}{2}, \infty\right) \setminus \{1\}$, we have that $f_0''(x) = \frac{\beta}{(1-\beta)^2}\left[\frac{1}{x^{\beta+1}}\left(1+x^{-\beta}\right)^{\frac{1-2\beta}{\beta}}\right] > 0$ for $x > 0$. Since $f_0'(1) = 0$, it follows that $f_0'(x)$ is increasing everywhere, negative on $(0,1)$ and positive on $(1,\infty)$. It follows that $g(x)$ is decreasing on $(0,1)$ and increasing on $(1,\infty)$. $\beta > 0$ means that $1 + x^{-\beta} \to 1$ as $x \to \infty$. Hence $g(x)$ is bounded above and increasing in $x$, thus $\lim_{x\to\infty} g(x)$ exists and is finite.

For $\beta \in \left(\frac{1}{2}, 1\right)$, $\frac{1-\beta}{\beta} > 0$. It follows that $\left(1+x^{-\beta}\right)^{\frac{1-\beta}{\beta}}$ grows unboundedly as $x \to 0$, and hence so does $g(x)$. Since $g(x)$ is decreasing on $(0,1)$, for any $0 < \delta < 1$ we have that $h_\delta(x) \geq g(x)$ on $(0,1)$. Since $g(x)$ is increasing on $(1,\infty)$ we have that $h_\delta(x) \geq \lim_{x\to\infty} g(x) \geq g(x)$ on $(1,\infty)$.

For $\beta \in (1,\infty)$, $\frac{1-\beta}{\beta} < 0$. It follows that $\left(1+x^{-\beta}\right)^{\frac{1-\beta}{\beta}} \to 0$ as $x \to 0$, and hence $\lim_{x\to 0} g(x)$ exists and is finite. Since $g(x)$ is decreasing on $(0,1)$ and increasing on $(1,\infty)$, it follows that $g(x) \leq \max\{\lim_{x\to 0} g(x), \lim_{x\to\infty} g(x)\}$ for all $x \in [0,\infty)$

$\square$

## B.4   Proof of Theorem 2

**Theorem 2** (Rates of the bias). *If $\mathbb{E}_{X\sim Q_X, Z\sim P_Z}\left[q^4(Z|X)/p^4(Z)\right]$ is finite then the bias $\mathbb{E}_{\mathbf{X}^N}\left[D_f(\hat{Q}_Z^N \| P_Z)\right] - D_f(Q_Z \| P_Z)$ decays with rate as given in the second row of Table 1.*

*Proof.* For each $f$-divergence we will work with the function $f_0$ which is decreasing on $(0,1)$ and increasing on $(1,\infty)$ with $D_f = D_{f_0}$ (see Appendix A).

For shorthand we will sometimes use the notation $\|q(z|X)/p(z)\|_{L_2(P_Z)}^2 = \int \frac{q(z|X)^2}{p(z)^2} p(z) dz$ and $\|q^2(z|X)/p^2(z)\|_{L_2(P_Z)}^2 = \int \frac{q(z|X)^4}{p(z)^4} p(z) dz$.

We will denote $C := \mathbb{E}_{X\sim Q_X, Z\sim P_Z}\left[q^4(Z|X)/p^4(Z)\right]$ which is finite by assumption. This implies that the second moment $B := \mathbb{E}_{X\sim Q_X, Z\sim P_Z}\left[q^2(Z|X)/p^2(Z)\right]$ is also finite, thanks to Jensen's inequality:

$$\mathbb{E}[Y^2] = \mathbb{E}[\sqrt{Y^4}] \leq \sqrt{\mathbb{E}[Y^4]}.$$

**The case that $D_f$ is the $\chi^2$-divergence:**   In this case, using $f(x) = x^2 - 1$, it can be seen that the bias is equal to

$$\mathop{\mathbb{E}}_{\mathbf{X}^N}\left[D_f\left(\hat{Q}_Z^N \| P_Z\right)\right] - D_f(Q_Z \| P_Z) = \mathop{\mathbb{E}}_{\mathbf{X}^N}\left[\int_Z \left(\frac{\hat{q}_N(z) - q(z)}{p(z)}\right)^2 dP(z)\right]. \tag{6}$$

Indeed, expanding the right hand side and using the fact that $\mathbb{E}_{\mathbf{X}^N}\, \hat{q}_N(z) = q(z)$ yields

$$\mathop{\mathbb{E}}_{\mathbf{X}^N}\left[\int_Z \frac{\hat{q}_N^2(z) - 2\hat{q}_N(z)q(z) + q^2(z)}{p^2(z)} dP(z)\right]$$

$$= \mathop{\mathbb{E}}_{\mathbf{X}^N}\left[\int_Z \frac{\hat{q}_N^2(z) - q^2(z)}{p^2(z)} dP(z)\right]$$

$$= \mathop{\mathbb{E}}_{\mathbf{X}^N}\left[\int_Z \left(\frac{\hat{q}_N^2(z)}{p^2(z)} - 1\right) dP(z)\right] - \int_Z \left(\frac{q^2(z)}{p^2(z)} - 1\right) dP(z)$$

$$= \mathop{\mathbb{E}}_{\mathbf{X}^N}\left[D_f\left(\hat{Q}_Z^N \| P_Z\right)\right] - D_f(Q_Z \| P_Z).$$

Again using the fact that $\mathbb{E}_{\mathbf{X}^N}\, \hat{q}_N(z) = q(z)$, observe that taking expectations over $\mathbf{X}^N$ in the right hand size of Equation 6 above (after changing the order of integration) can be viewed as taking the variance of $\hat{q}_N(z)/p(z)$, the average of $N$ i.i.d. random variables, and so

$$\mathop{\mathbb{E}}_{\mathbf{X}^N}\left[\int_Z \left(\frac{\hat{q}_N(z) - q(z)}{p(z)}\right)^2 dP(z)\right] = \int_Z \mathop{\mathbb{E}}_{\mathbf{X}^N}\left[\left(\frac{\hat{q}_N(z) - q(z)}{p(z)}\right)^2\right] dP(z)$$

$$= \frac{1}{N} \int_Z \mathop{\mathbb{E}}_X\left[\left(\frac{q(z|X) - q(z)}{p(z)}\right)^2\right] dP(z)$$

$$= \frac{1}{N} \mathop{\mathbb{E}}_X \chi^2\left(Q_{Z|X} \| P_Z\right) - \frac{1}{N}\chi^2\left(Q_Z \| P_Z\right)$$

$$\leq \frac{B-1}{N}.$$

**The case that $D_f$ is the Total Variation distance or $D_{f_\beta}$ with $\beta > 1$:** For these divergences, we only need the condition that the second moment $\mathbb{E}_X \|q(z|X)/p(z)\|_{L_2(P_Z)}^2 < \infty$ is bounded.

$$\mathbb{E}_{\mathbf{X}^N} \left[ D_{f_0} \left( \hat{Q}_Z^N \| P_Z \right) \right] - D_{f_0} (Q_Z \| P_Z)$$

$$= \mathbb{E}_{\mathbf{X}^N} \mathbb{E}_{P_Z} \left[ f_0 \left( \frac{\hat{q}_N(z)}{p(z)} \right) - f_0 \left( \frac{q(z)}{p(z)} \right) \right]$$

$$\leq \mathbb{E}_{\mathbf{X}^N} \mathbb{E}_{P_Z} \left[ \left( \frac{\hat{q}_N(z) - q(z)}{p(z)} \right) f_0' \left( \frac{\hat{q}_N(z)}{p(z)} \right) \right]$$

$$\leq \underbrace{\sqrt{\mathbb{E}_{\mathbf{X}^N} \mathbb{E}_{P_Z} \left[ \left( \frac{\hat{q}_N(z) - q(z)}{p(z)} \right)^2 \right]}}_{(i)} \times \underbrace{\sqrt{\mathbb{E}_{\mathbf{X}^N} \mathbb{E}_{P_Z} \left[ f_0'^2 \left( \frac{\hat{q}_N(z)}{p(z)} \right) \right]}}_{(ii)}$$

where the first inequality holds due to convexity of $f_0$ and the second inequality follows by Cauchy-Schwartz. Then,

$$(i)^2 = \mathbb{E}_{P_Z} \text{Var}_{\mathbf{X}^N} \left[ \frac{\hat{q}_N(z)}{p(z)} \right]$$

$$= \frac{1}{N} \mathbb{E}_{P_Z} \text{Var}_X \left[ \frac{q(z|X)}{p(z)} \right]$$

$$\leq \frac{1}{N} \mathbb{E}_X \mathbb{E}_{P_Z} \left[ \frac{q^2(z|X)}{p^2(z)} \right] = \frac{1}{N} \mathbb{E}_X \left\| \frac{q(z|X)}{p(z)} \right\|_{L_2(P_Z)}^2$$

$$\implies (i) = O \left( \frac{1}{\sqrt{N}} \right).$$

For Total Variation, $f_0'^2(x) \leq 1$, so

$$(ii)^2 \leq 1.$$

For $D_{f_\beta}$ with $\beta > 1$, Lemma 6 shows that $f_0'^2(x) \leq \max\{\lim_{x \to 0} f_0'^2(x), \lim_{x \to \infty} f_0'^2(x)\} < \infty$ and so

$$(ii)^2 = O(1).$$

Thus, for both cases considered,

$$\mathbb{E}_{\mathbf{X}^N} \left[ D_f \left( \hat{Q}_Z^N \| P_Z \right) \right] - D_f (Q_Z \| P_Z) \leq O \left( \frac{1}{\sqrt{N}} \right).$$

**All other divergences.** We start by writing the difference as the sum of integrals over mutually exclusive events that partition $\mathcal{Z}$. Denoting by $\gamma_N$ and $\delta_N$ scalars depending on $N$, write

$$\mathbb{E}_{\mathbf{X}^N} \left[ D_f \left( \hat{Q}_Z^N \| P_Z \right) \right] - D_f (Q_Z \| P_Z)$$

$$= \mathbb{E}_{\mathbf{X}^N} \left[ \int f_0 \left( \frac{\hat{q}_N(z)}{p(z)} \right) - f_0 \left( \frac{q(z)}{p(z)} \right) dP_Z(z) \right]$$

$$= \mathbb{E}_{\mathbf{X}^N} \left[ \int f_0 \left( \frac{\hat{q}_N(z)}{p(z)} \right) - f_0 \left( \frac{q(z)}{p(z)} \right) \mathbb{1}_{\left\{ \frac{\hat{q}_N(z)}{p(z)} \leq \delta_N \text{ and } \frac{q(z)}{p(z)} \leq \gamma_N \right\}} dP_Z(z) \right] \quad \text{(A)}$$

$$+ \mathbb{E}_{\mathbf{X}^N} \left[ \int f_0 \left( \frac{\hat{q}_N(z)}{p(z)} \right) - f_0 \left( \frac{q(z)}{p(z)} \right) \mathbb{1}_{\left\{ \frac{\hat{q}_N(z)}{p(z)} \leq \delta_N \text{ and } \frac{q(z)}{p(z)} > \gamma_N \right\}} dP_Z(z) \right] \quad \text{(B)}$$

$$+ \mathbb{E}_{\mathbf{X}^N} \left[ \int f_0 \left( \frac{\hat{q}_N(z)}{p(z)} \right) - f_0 \left( \frac{q(z)}{p(z)} \right) \mathbb{1}_{\left\{ \frac{\hat{q}_N(z)}{p(z)} > \delta_N \right\}} dP_Z(z) \right]. \quad \text{(C)}$$

Consider each of the terms (A), (B) and (C) separately.

Later on, we will pick $\delta_N < \gamma_N$ to be decreasing in $N$. In the worst case, $N > 8$ will be sufficient to ensure that $\gamma_N < 1$, so in the remainder of this proof we will assume that $\delta_N, \gamma_N < 1$.

$\widehat{A}$: Recall that $f_0(x)$ is decreasing on the interval $[0, 1]$. Since $\gamma_N, \delta_N \leq 1$, the integrand is at most $f_0(0) - f_0(\gamma_N)$, and so

$$\widehat{A} \leq f_0(0) - f_0(\gamma_N).$$

$\widehat{B}$: The integrand is bounded above by $f_0(0)$ since $\delta_N < 1$, and so

$$\widehat{B} \leq f_0(0) \times \underbrace{\mathbb{P}_{Z,\mathbf{X}^N} \left\{ \frac{\hat{q}_N(z)}{p(z)} \leq \delta_N \text{ and } \frac{q(z)}{p(z)} > \gamma_N \right\}}_{\circledast}.$$

We will upper bound $\mathbb{P}_{Z,\mathbf{X}^N}\circledast$: observe that if $\gamma_N > \delta_N$, then $\circledast \implies \left| \frac{\hat{q}_N(z) - q(z)}{p(z)} \right| \geq \gamma_N - \delta_N$. It thus follows that

$$\begin{aligned}
\mathbb{P}_{Z,\mathbf{X}^N}\circledast &\leq \mathbb{P}_{Z,\mathbf{X}^N} \left\{ \left| \frac{\hat{q}_N(z) - q(z)}{p(z)} \right| \geq \gamma_N - \delta_N \right\} \\
&= \mathbb{E}_Z \left[ \mathbb{P}_{\mathbf{X}^N} \left\{ \left| \frac{\hat{q}_N(z) - q(z)}{p(z)} \right| \geq \gamma_N - \delta_N \mid Z \right\} \right] \\
&\leq \mathbb{E}_Z \left[ \frac{\text{Var}_{\mathbf{X}^N} \left[ \frac{\hat{q}_N(z)}{p(z)} \right]}{(\gamma_N - \delta_N)^2} \right] \\
&= \frac{1}{N(\gamma_N - \delta_N)^2} \mathbb{E}_Z \left[ \mathbb{E}_X \left[ \frac{q^2(z|X)}{p^2(z)} \right] - \frac{q^2(z)}{p^2(z)} \right] \\
&\leq \frac{1}{N(\gamma_N - \delta_N)^2} \mathbb{E}_Z \mathbb{E}_X \left[ \frac{q^2(z|X)}{p^2(z)} \right] \\
&\leq \frac{\sqrt{C}}{N(\gamma_N - \delta_N)^2}.
\end{aligned}$$

The second inequality follows by Chebyshev's inequality, noting that $\mathbb{E}_{\mathbf{X}^N} \frac{\hat{q}_N(z)}{p(z)} = \frac{q(z)}{p(z)}$. The penultimate inequality is due to dropping a negative term. The final inequality is due to the boundedness assumption $C = \mathbb{E}_X \left\| \frac{q^2(z|X)}{p^2(z)} \right\|_{L_2(P_Z)}^2$. We thus have that

$$\widehat{B} \leq f_0(0) \frac{\sqrt{C}}{N(\gamma_N - \delta_N)^2}.$$

$\widehat{C}$: Bounding this term will involve two computations, one of which (††) will be treated separately for each divergence we consider.

$$\begin{aligned}
\widehat{C} &= \mathbb{E}_{\mathbf{X}^N} \left[ \int f_0 \left( \frac{\hat{q}_N(z)}{p(z)} \right) - f_0 \left( \frac{q(z)}{p(z)} \right) \mathbb{1}_{\left\{ \frac{\hat{q}_N(z)}{p(z)} > \delta_N \right\}} dP_Z(z) \right] \\
&\leq \mathbb{E}_{\mathbf{X}^N} \left[ \int \left( \frac{\hat{q}_N(z)}{p(z)} - \frac{q(z)}{p(z)} \right) f_0' \left( \frac{\hat{q}_N(z)}{p(z)} \right) \mathbb{1}_{\left\{ \frac{\hat{q}_N(z)}{p(z)} > \delta_N \right\}} dP_Z(z) \right] \quad \text{(Convexity of } f) \\
&\leq \underbrace{\sqrt{ \mathbb{E}_{\mathbf{X}^N} \mathbb{E}_Z \left[ \left( \frac{\hat{q}_N(z)}{p(z)} - \frac{q(z)}{p(z)} \right)^2 \right] }}_{(\dagger)} \times \underbrace{\sqrt{ \mathbb{E}_{\mathbf{X}^N} \mathbb{E}_Z \left[ f_0'^2 \left( \frac{\hat{q}_N(z)}{p(z)} \right) \mathbb{1}_{\left\{ \frac{\hat{q}_N(z)}{p(z)} > \delta_N \right\}} \right] }}_{(\dagger\dagger)} \quad \text{(Cauchy-Schwartz)}
\end{aligned}$$

Noting that $\mathbb{E}_X \frac{q(z|X)}{p(z)} = \frac{q(z)}{p(z)}$, we have that

$$
\begin{aligned}
(\dagger)^2 &= \underset{Z}{\mathbb{E}}\, \mathrm{Var}_{\mathbf{X}^N}\left[\frac{\hat{q}_N(z)}{p(z)}\right] \\
&= \frac{1}{N}\underset{Z}{\mathbb{E}}\, \mathrm{Var}_X\left[\frac{q(z|X)}{p(z)}\right] \\
&\leq \frac{1}{N}\underset{X}{\mathbb{E}}\left\|\frac{q(z|X)}{p(z)}\right\|^2_{L_2(P_Z)} \\
\Longrightarrow (\dagger) &\leq \frac{\sqrt{B}}{\sqrt{N}}
\end{aligned}
$$

where $\sqrt{B} = \sqrt{\mathbb{E}_X\left\|\frac{q(z|X)}{p(z)}\right\|^2_{L_2(P_Z)}}$ is finite by assumption.

Term ($\dagger\dagger$) will be bounded differently for each divergence, though using a similar pattern. The idea is to use the results of Lemmas 2–6 in order to upper bound $f_0'^2(x)$ with something that can be easily integrated.

**KL.** By Lemma 2, there exists a function $h_{\delta_N}(x)$ that is positive and concave on $[0,\infty)$ and is an upper bound of $f_0'^2(x)$ on $[\delta_N, \infty)$ with $h_{\delta_N}(1) = \log^2(\delta_N) + \frac{2}{e}$.

$$
\begin{aligned}
(\dagger\dagger)^2 &= \underset{\mathbf{X}^N}{\mathbb{E}}\left[\int f_0'^2\left(\frac{\hat{q}_N(z)}{p(z)}\right)\mathbb{1}_{\left\{\frac{\hat{q}_N(z)}{p(z)}>\delta_N\right\}}p(z)dz\right] \\
&\leq \underset{\mathbf{X}^N}{\mathbb{E}}\left[\int h_{\delta_N}\left(\frac{\hat{q}_N(z)}{p(z)}\right)\mathbb{1}_{\left\{\frac{\hat{q}_N(z)}{p(z)}>\delta_N\right\}}p(z)dz\right] && (h_{\delta_N}\text{ upper bounds } f'^2 \text{ on } (\delta_N,\infty)) \\
&\leq \underset{\mathbf{X}^N}{\mathbb{E}}\left[\int h_{\delta_N}\left(\frac{\hat{q}_N(z)}{p(z)}\right)p(z)dz\right] && (h_{\delta_N}\text{ non-negative on } [0,\infty)) \\
&\leq \underset{\mathbf{X}^N}{\mathbb{E}}\left[h_{\delta_N}\left(\int \frac{\hat{q}_N(z)}{p(z)}p(z)dz\right)\right] && (h_{\delta_N}\text{ concave}) \\
&= h_{\delta_N}(1) \\
&= \log^2(\delta_N) + \frac{2}{e} \\
\Longrightarrow (\dagger\dagger) &= \sqrt{\log^2(\delta_N) + \frac{2}{e}}.
\end{aligned}
$$

Therefore,

$$
\text{\textcircled{C}} \leq \sqrt{B}\sqrt{\frac{\log^2(\delta_N) + \frac{2}{e}}{N}}.
$$

Putting everything together,

$$
\begin{aligned}
&\underset{\mathbf{X}^N}{\mathbb{E}}\left[D_f\left(\hat{Q}_Z^N\|P_Z\right)\right] - D_f(Q_Z\|P_Z) \\
&\leq \text{\textcircled{A}} + \text{\textcircled{B}} + \text{\textcircled{C}} \\
&\leq f_0(0) - f_0(\gamma_N) + f_0(0)\frac{\sqrt{C}}{N(\gamma_N - \delta_N)^2} + \sqrt{B}\sqrt{\frac{\log^2(\delta_N) + \frac{2}{e}}{N}} \\
&= \gamma_N - \gamma_N\log\gamma_N + \frac{\sqrt{C}}{N(\gamma_N - \delta_N)^2} + \sqrt{B}\sqrt{\frac{\log^2(\delta_N) + \frac{2}{e}}{N}}.
\end{aligned}
$$

Taking $\delta_N = \frac{1}{N^{1/3}}$ and $\gamma_N = \frac{2}{N^{1/3}}$:

$$= \frac{2}{N^{1/3}} - \frac{2}{N^{1/3}} \log\left(\frac{2}{N^{1/3}}\right) + \frac{\sqrt{C}}{N \cdot \frac{1}{N^{2/3}}} + \sqrt{B}\sqrt{\frac{\log^2\left(\frac{1}{N^{1/3}}\right) + \frac{2}{e}}{N}}$$

$$= \frac{2 - 2\log 2}{N^{1/3}} + \frac{2}{3}\frac{\log N}{N^{1/3}} + \frac{\sqrt{C}}{N^{1/3}} + \sqrt{B}\sqrt{\frac{\frac{1}{4}\log^2(N) + \frac{2}{e}}{N}}$$

$$= O\left(\frac{\log N}{N^{1/3}}\right)$$

**Squared-Hellinger.** Lemma 3 provides a function $h_\delta$ that upper bounds $f'^2(x)$ for $x \in\in [\delta, \infty)$.

$$(\dagger\dagger)^2 = \mathop{\mathbb{E}}_{\mathbf{X}^N}\left[\int f_0'^2\left(\frac{\hat{q}_N(z)}{p(z)}\right)\mathbb{1}_{\left\{\frac{\hat{q}_N(z)}{p(z)} > \delta_N\right\}} p(z)dz\right]$$

$$\leq \mathop{\mathbb{E}}_{\mathbf{X}^N}\left[\int h_{\delta_N}\left(\frac{\hat{q}_N(z)}{p(z)}\right)\mathbb{1}_{\left\{\frac{\hat{q}_N(z)}{p(z)} > \delta_N\right\}} p(z)dz\right] \quad (h_{\delta_N} \text{ upper bounds } f_0'^2 \text{ on } (\delta_N, \infty))$$

$$\leq \mathop{\mathbb{E}}_{\mathbf{X}^N}\left[\int h_{\delta_N}\left(\frac{\hat{q}_N(z)}{p(z)}\right) p(z)dz\right] \quad (h_{\delta_N} \text{ non-negative on } [0, \infty))$$

$$= \frac{1}{\delta_N}\mathop{\mathbb{E}}_{\mathbf{X}^N}\mathop{\mathbb{E}}_{P_Z}\left[\left(\frac{\hat{q}_N(z)}{p(z)} - 1\right)^2\right]$$

$$\leq \frac{1}{\delta_N}\mathop{\mathbb{E}}_{\mathbf{X}^N}\mathop{\mathbb{E}}_{P_Z}\left[\left(\frac{\hat{q}_N(z)}{p(z)}\right)^2 + 1\right]$$

$$= \frac{1}{\delta_N} + \frac{1}{\delta_N}\mathop{\mathbb{E}}_{\mathbf{X}^N}\left[\left\|\frac{\hat{q}_N(z)}{p(z)}\right\|_{L_2(P_Z)}^2\right]$$

$$\leq \frac{B+1}{\delta_N}$$

$$\implies (\dagger\dagger) = \frac{\sqrt{B+1}}{\sqrt{\delta_N}}.$$

and thus

$$\mathop{\mathbb{E}}_{\mathbf{X}^N}\left[D_f\left(\hat{Q}_Z^N \| P_Z\right)\right] - D_f(Q_Z \| P_Z)$$

$$\leq \text{Ⓐ} + \text{Ⓑ} + \text{Ⓒ}$$

$$\leq f_0(0) - f_0(\gamma_N) + f_0(0)\frac{\sqrt{C}}{N(\gamma_N - \delta_N)^2} + \frac{\sqrt{B}\sqrt{B+1}}{\sqrt{N\delta_N}}$$

$$= 2\sqrt{\gamma_N} + \frac{2\sqrt{C}}{N(\gamma_N - \delta_N)^2} + \frac{\sqrt{B}\sqrt{B+1}}{\sqrt{N\delta_N}}.$$

Setting $\gamma_N = \frac{2}{N^{2/5}}$ and $\delta_N = \frac{1}{N^{2/5}}$ yields

$$= \frac{2}{N^{1/5}} + \frac{2\sqrt{C}}{N^{1/5}} + \frac{\sqrt{B}\sqrt{B+1}}{N^{3/10}}$$

$$= O\left(\frac{1}{N^{1/5}}\right)$$

$\alpha$-**divergence with** $\alpha \in (-1, 1)$**.** Lemma 4 provides a function $h_\delta$ that upper bounds $f'^2(x)$ for $x \in\in [\delta, \infty)$.

$$
\begin{aligned}
(\dagger\dagger)^2 &= \underset{\mathbf{X}^N}{\mathbb{E}} \left[ \int f_0'^2 \left( \frac{\hat{q}_N(z)}{p(z)} \right) \mathbb{1}_{\left\{ \frac{\hat{q}_N(z)}{p(z)} > \delta_N \right\}} p(z) dz \right] \\
&\leq \underset{\mathbf{X}^N}{\mathbb{E}} \left[ \int h_{\delta_N} \left( \frac{\hat{q}_N(z)}{p(z)} \right) \mathbb{1}_{\left\{ \frac{\hat{q}_N(z)}{p(z)} > \delta_N \right\}} p(z) dz \right] \qquad (h_{\delta_N} \text{ upper bounds } f_0'^2 \text{ on } (\delta_N, \infty)) \\
&\leq \underset{\mathbf{X}^N}{\mathbb{E}} \left[ \int h_{\delta_N} \left( \frac{\hat{q}_N(z)}{p(z)} \right) p(z) dz \right] \qquad (h_{\delta_N} \text{ non-negative on } [0, \infty)) \\
&= \frac{4 \left( \delta_N^{\frac{\alpha-1}{2}} - 1 \right)^2}{(\alpha-1)^2 (\delta_N - 1)^2} \underset{\mathbf{X}^N}{\mathbb{E}} \underset{P_Z}{\mathbb{E}} \left[ \left( \frac{\hat{q}_N(z)}{p(z)} - 1 \right)^2 \right] \\
&\leq \frac{4 \left( \delta_N^{\frac{\alpha-1}{2}} - 1 \right)^2}{(\alpha-1)^2 (\delta_N - 1)^2} \underset{\mathbf{X}^N}{\mathbb{E}} \underset{P_Z}{\mathbb{E}} \left[ \left( \frac{\hat{q}_N(z)}{p(z)} \right)^2 + 1 \right] \\
&= \frac{4 \left( \delta_N^{\frac{\alpha-1}{2}} - 1 \right)^2}{(\alpha-1)^2 (\delta_N - 1)^2} \left( 1 + \underset{\mathbf{X}^N}{\mathbb{E}} \left[ \left\| \frac{\hat{q}_N(z)}{p(z)} \right\|_{L_2(P_Z)}^2 \right] \right) \\
&\leq \frac{4(1+B) \left( \delta_N^{\frac{\alpha-1}{2}} - 1 \right)^2}{(\alpha-1)^2 (\delta_N - 1)^2} \\
\implies (\dagger\dagger) &= \frac{2\sqrt{1+B} \left( \delta_N^{\frac{\alpha-1}{2}} - 1 \right)}{(\alpha-1)(\delta_N - 1)}.
\end{aligned}
$$

and thus

$$
\begin{aligned}
&\underset{\mathbf{X}^N}{\mathbb{E}} \left[ D_f \left( \hat{Q}_Z^N \| P_Z \right) \right] - D_f \left( Q_Z \| P_Z \right) \\
&\leq \text{Ⓐ} + \text{Ⓑ} + \text{Ⓒ} \\
&\leq f_0(0) - f_0(\gamma_N) + f_0(0) \frac{\sqrt{C}}{N (\gamma_N - \delta_N)^2} + \frac{2\sqrt{B}\sqrt{1+B} \left( \delta_N^{\frac{\alpha-1}{2}} - 1 \right)}{(\alpha-1)(\delta_N - 1)\sqrt{N}} \\
&\leq k_1 \gamma_N^{\frac{\alpha+1}{2}} + k_2 \gamma_N + \frac{k_3}{N(\gamma_N - \delta_N)^2} + \frac{k_4 \delta_N^{\frac{\alpha-1}{2}}}{\sqrt{N}}.
\end{aligned}
$$

where each $k_i$ is a positive constant independent of $N$.

Setting $\gamma_N = \frac{2}{N^{\frac{2}{\alpha+5}}}$ and $\delta_N = \frac{1}{N^{\frac{2}{\alpha+5}}}$ yields

$$
\begin{aligned}
&=\leq \frac{k_1}{N^{\frac{\alpha+1}{\alpha+5}}} + \frac{k_2}{N^{\frac{2}{\alpha+5}}} + \frac{k_3}{N^{\frac{\alpha+1}{\alpha+5}}} + \frac{k_4}{N^{\frac{7-\alpha}{2(\alpha+5)}}} \\
&= O \left( \frac{1}{N^{\frac{\alpha+1}{\alpha+5}}} \right)
\end{aligned}
$$

**Jensen-Shannon.** Lemma 5 provides a function $h_\delta$ that upper bounds $f'^2(x)$ for $x \in [\delta, \infty)$.

$$(\dagger\dagger)^2 = \mathbb{E}_{\mathbf{x}^N}\left[\int f_0'^2\left(\frac{\hat{q}_N(z)}{p(z)}\right)\mathbb{1}_{\left\{\frac{\hat{q}_N(z)}{p(z)}>\delta_N\right\}}p(z)dz\right]$$

$$\leq \mathbb{E}_{\mathbf{x}^N}\left[\int h_{\delta_N}\left(\frac{\hat{q}_N(z)}{p(z)}\right)\mathbb{1}_{\left\{\frac{\hat{q}_N(z)}{p(z)}>\delta_N\right\}}p(z)dz\right] \qquad (h_{\delta_N} \text{ upper bounds } f_0'^2 \text{ on } (\delta_N,\infty))$$

$$\leq \mathbb{E}_{\mathbf{x}^N}\left[\int h_{\delta_N}\left(\frac{\hat{q}_N(z)}{p(z)}\right)p(z)dz\right] \qquad (h_{\delta_N} \text{ non-negative on } [0,\infty))$$

$$= 5\log^2 2 + \log^2\left(\frac{\delta_N}{1+\delta_N}\right) + 2\log 2\log\left(\frac{\delta_N}{1+\delta_N}\right)$$

$$= 5\log^2 2 + \log^2\left(1+\frac{1}{\delta_N}\right) - 2\log 2\log\left(1+\frac{1}{\delta_N}\right)$$

$$\leq 5\log^2 2 + 5\log^2\left(1+\frac{1}{\delta_N}\right) + 10\log 2\log\left(1+\frac{1}{\delta_N}\right)$$

$$= 5\left(\log\left(1+\frac{1}{\delta_N}\right) - \log 2\right)^2$$

$$\implies (\dagger\dagger) \leq \sqrt{5}\log\left(1+\frac{1}{\delta_N}\right) - \sqrt{5}\log 2$$

$$\leq \sqrt{5}\log\left(\frac{2}{\delta_N}\right) - \sqrt{5}\log 2 \qquad (\text{since } \delta_N < 1)$$

$$= -\sqrt{5}\log(\delta_N).$$

and thus

$$\mathbb{E}_{\mathbf{x}^N}\left[D_f\left(\hat{Q}_Z^N\|P_Z\right)\right] - D_f\left(Q_Z\|P_Z\right)$$

$$\leq \text{\textcircled{A}} + \text{\textcircled{B}} + \text{\textcircled{C}}$$

$$\leq f_0(0) - f_0(\gamma_N) + f_0(0)\frac{\sqrt{C}}{N(\gamma_N - \delta_N)^2} - \frac{\sqrt{5}\sqrt{B}\log\delta_N}{\sqrt{N}}$$

$$\leq \gamma_N\log\left(\frac{1+\gamma_N}{2\gamma_N}\right) + \log(1+\gamma_N) + \frac{\log 2\sqrt{C}}{N(\gamma_N - \delta_N)^2} - \frac{\sqrt{5}\sqrt{B}\log\delta_N}{\sqrt{N}}$$

Using the fact that $\gamma_N\log(1+\gamma_N) \leq \gamma_N\log 2$ for $\gamma_N < 1$ and $\log(1+\gamma_N) \leq \gamma_N$, we can upper bound the last line with

$$\leq \gamma_N(\log 2 + 1) - \gamma_N\log\gamma_N + \frac{\log 2\sqrt{C}}{N(\gamma_N - \delta_N)^2} - \frac{\sqrt{5}\sqrt{B}\log\delta_N}{\sqrt{N}}$$

Setting $\gamma_N = \frac{2}{N^{\frac{1}{3}}}$ and $\delta_N = \frac{1}{N^{\frac{1}{3}}}$ yields

$$= \frac{k_1}{N^{\frac{1}{3}}} + \frac{k_2\log N}{N^{\frac{1}{3}}} + \frac{k_3}{N^{\frac{1}{3}}} + \frac{k_4\log N}{N^{\frac{1}{2}}}$$

$$= O\left(\frac{\log N}{N^{\frac{1}{3}}}\right)$$

where the $k_i$ are positive constants independent of $N$.

**$f_\beta$-divergence with $\beta \in (\frac{1}{2}, 1)$.** Lemma 6 provides a function $h_\delta$ that upper bounds $f'^2(x)$ for $x \in [\delta, \infty)$.

$$(\dagger\dagger)^2 = \mathop{\mathbb{E}}_{\mathbf{X}^N}\left[\int f_0'^2\left(\frac{\hat{q}_N(z)}{p(z)}\right)\mathbb{1}_{\left\{\frac{\hat{q}_N(z)}{p(z)}>\delta_N\right\}}p(z)dz\right]$$

$$\leq \mathop{\mathbb{E}}_{\mathbf{X}^N}\left[\int h_{\delta_N}\left(\frac{\hat{q}_N(z)}{p(z)}\right)\mathbb{1}_{\left\{\frac{\hat{q}_N(z)}{p(z)}>\delta_N\right\}}p(z)dz\right] \qquad (h_{\delta_N} \text{ upper bounds } f_0'^2 \text{ on } (\delta_N,\infty))$$

$$\leq \mathop{\mathbb{E}}_{\mathbf{X}^N}\left[\int h_{\delta_N}\left(\frac{\hat{q}_N(z)}{p(z)}\right)p(z)dz\right] \qquad (h_{\delta_N} \text{ non-negative on } [0,\infty))$$

$$= \left(\frac{\beta}{1-\beta}\right)^2\left[\left(1+\delta_N^{-\beta}\right)^{\frac{1-\beta}{\beta}}-2^{\frac{1-\beta}{\beta}}\right]^2 + \frac{\beta^2}{(1-\beta)^2}\left(2^{\frac{1-\beta}{\beta}}\right)^2$$

$$\leq 2\left(\frac{\beta}{1-\beta}\right)^2\left[\left(1+\delta_N^{-\beta}\right)^{\frac{1-\beta}{\beta}}+2^{\frac{1-\beta}{\beta}}\right]^2$$

$$\leq 2\left(\frac{\beta}{1-\beta}\right)^2\left[2\left(2\delta_N^{-\beta}\right)^{\frac{1-\beta}{\beta}}\right]^2 \qquad (\text{since } \delta_N < 1 \text{ and } \beta > 0 \text{ implies } \delta_N^{-\beta} > 1)$$

$$= 2^{\frac{2+\beta}{\beta}}\left(\frac{\beta}{1-\beta}\right)^2\delta_N^{2(\beta-1)}$$

$$\implies (\dagger\dagger) \leq 2^{\frac{2+\beta}{2\beta}}\left(\frac{\beta}{1-\beta}\right)\delta_N^{\beta-1}$$

(noting that $\frac{\beta^2}{(1-\beta)^2}\left(2^{\frac{1}{\beta}-1}\right)^2 = \lim_{x\to\infty}f_0'^2(x)$ as defined in Lemma 6). Thus

$$\mathop{\mathbb{E}}_{\mathbf{X}^N}\left[D_f\left(\hat{Q}_Z^N\|P_Z\right)\right] - D_f\left(Q_Z\|P_Z\right)$$

$$\leq \textcircled{A} + \textcircled{B} + \textcircled{C}$$

$$\leq f_0(0) - f_0(\gamma_N) + f_0(0)\frac{\sqrt{C}}{N(\gamma_N-\delta_N)^2} + \frac{\sqrt{B}}{\sqrt{N}}2^{\frac{2+\beta}{2\beta}}\left(\frac{\beta}{1-\beta}\right)\delta_N^{\beta-1}$$

$$\leq \frac{\beta}{1-\beta}\left[1-\left(1+\delta_N^\beta\right)^{1/\beta}+2^{\frac{1-\beta}{\beta}}\delta_N\right] + f_0(0)\frac{\sqrt{C}}{N(\gamma_N-\delta_N)^2} + \frac{\sqrt{B}}{\sqrt{N}}2^{\frac{2+\beta}{2\beta}}\left(\frac{\beta}{1-\beta}\right)\delta_N^{\beta-1}$$

$$\leq \frac{\beta}{1-\beta}2^{\frac{1-\beta}{\beta}}\delta_N + f_0(0)\frac{\sqrt{C}}{N(\gamma_N-\delta_N)^2} + \frac{\sqrt{B}}{\sqrt{N}}2^{\frac{2+\beta}{2\beta}}\left(\frac{\beta}{1-\beta}\right)\delta_N^{\beta-1}$$

$$= k_1\delta_N + \frac{k_2}{N(\gamma_N-\delta_N)^2} + \frac{k_3\delta_N^{\beta-1}}{\sqrt{N}}$$

where the $k_i$ are positive constants independent of $N$.

Setting $\gamma_N = \frac{2}{N^{\frac{1}{3}}}$ and $\delta_N = \frac{1}{N^{\frac{1}{3}}}$ yields

$$= \frac{k_1}{N^{\frac{1}{3}}} + \frac{k_2}{N^{\frac{1}{3}}} + \frac{k_3}{N^{\frac{1}{2}+\frac{\beta-1}{3}}}$$

$$= O\left(\frac{1}{N^{\frac{1}{3}}}\right)$$

$\square$

## B.5  Proof of Theorem 3

We will make use of McDiarmid's theorem in our proof of Theorem 3:

**Theorem** (McDiarmid's inequality). *Suppose that $X_1,\dots,X_N \in \mathcal{X}$ are independent random variables and that $\phi : \mathcal{X}^N \to \mathbb{R}$ is a function. If it holds that for all $i \in \{1,\dots,N\}$ and $x_1,\dots,x_N,x_{i'}$,*

$$|\phi(x_1,\dots,x_{i-1},x_i,x_{i+1},\dots,x_N) - \phi(x_1,\dots,x_{i-1},x_{i'},x_{i+1},\dots,x_N)| \leq c_i,$$

*then*

$$\mathbb{P}\left(\phi(X_1,\ldots,X_N) - \mathbb{E}\,\phi \geq t\right) \leq \exp\left(\frac{-2t^2}{\sum_{i=1}^N c_i^2}\right)$$

*and*

$$\mathbb{P}\left(\phi(X_1,\ldots,X_N) - \mathbb{E}\,\phi \geq -t\right) \leq \exp\left(\frac{-2t^2}{\sum_{i=1}^N c_i^2}\right)$$

In our setting we will consider $\phi(\mathbf{X}^N) = D_f\left(\hat{Q}_Z^N \| P_Z\right)$.

**Theorem 3** (Tail bounds for RAM). *Suppose that $\chi^2\left(Q_{Z|x} \| P_Z\right) \leq C < \infty$ for all $x$ and for some constant $C$. Then, the RAM estimator $D_f(\hat{Q}_Z^N \| P_Z)$ concentrates to its mean in the following sense. For $N > 8$ and for any $\delta > 0$, with probability at least $1 - \delta$ it holds that*

$$\left| D_f(\hat{Q}_Z^N \| P_Z) - \mathbb{E}_{\mathbf{X}^N}\left[D_f(\hat{Q}_Z^N \| P_Z)\right]\right| \leq K \cdot \psi(N)\,\sqrt{\log(2/\delta)},$$

*where $K$ is a constant and $\psi(N)$ is given in Table 2.*

*Proof (Theorem 3).* We will show that $D_f\left(\hat{Q}_Z^N \| P_Z\right)$ exhibits the bounded difference property as in the statement of McDiarmid's theorem. Since $\hat{q}_N(z)$ is symmetric in the indices of $\mathbf{X}^N$, we can without loss of generality consider only the case $i = 1$. Henceforth, suppose $\mathbf{X}^N, \mathbf{X}^{N'}$ are two batches of data with $\mathbf{X}_1^N \neq \mathbf{X}_1^{N'}$ and $\mathbf{X}_i^N = \mathbf{X}_i^{N'}$ for all $i > 1$. For the remainder of this proof we will write explicitly the dependence of $\hat{Q}_Z^N$ on $\mathbf{X}^N$. We will write $\hat{Q}_Z^N(\mathbf{X}^N)$ for the probability measure and $\hat{q}_N(z; \mathbf{X}^N)$ for its density.

We will show that $\left| D_f\left(\hat{Q}_Z^N(\mathbf{X}^N) \| P_Z\right) - D_f\left(\hat{Q}_Z^N(\mathbf{X}^{N'}) \| P_Z\right)\right| \leq c_N$ where $c_N$ is a constant depending only on $N$. From this fact, McDiarmid's theorem and the union bound, it follows that:

$$\mathbb{P}\left(\left| D_f\left(\hat{Q}_Z^N(\mathbf{X}^N) \| P_Z\right) - \mathbb{E}_{\mathbf{X}^N} D_f\left(\hat{Q}_Z^N(\mathbf{X}^N) \| P_Z\right)\right| \geq t\right)$$

$$= \mathbb{P}\left(D_f\left(\hat{Q}_Z^N(\mathbf{X}^N) \| P_Z\right) - \mathbb{E}_{\mathbf{X}^N} D_f\left(\hat{Q}_Z^N(\mathbf{X}^N) \| P_Z\right) \geq t \text{ or}\right.$$

$$\left. D_f\left(\hat{Q}_Z^N(\mathbf{X}^N) \| P_Z\right) - \mathbb{E}_{\mathbf{X}^N} D_f\left(\hat{Q}_Z^N(\mathbf{X}^N) \| P_Z\right) \leq -t\right)$$

$$\leq \mathbb{P}\left(D_f\left(\hat{Q}_Z^N(\mathbf{X}^N) \| P_Z\right) - \mathbb{E}_{\mathbf{X}^N} D_f\left(\hat{Q}_Z^N(\mathbf{X}^N) \| P_Z\right) \geq t\right) +$$

$$\mathbb{P}\left(D_f\left(\hat{Q}_Z^N(\mathbf{X}^N) \| P_Z\right) - \mathbb{E}_{\mathbf{X}^N} D_f\left(\hat{Q}_Z^N(\mathbf{X}^N) \| P_Z\right) \leq -t\right)$$

$$\leq 2\exp\left(\frac{-2t^2}{Nc_N^2}\right).$$

Observe that by setting $t = \sqrt{\frac{Nc_N^2}{2}\log\left(\frac{2}{\delta}\right)}$,

the above inequality is equivalent to the statement that for any $\delta > 0$, with probability at least $1 - \delta$

$$\left| D_f\left(\hat{Q}_Z^N(\mathbf{X}^N) \| P_Z\right) - \mathbb{E}_{\mathbf{X}^N} D_f\left(\hat{Q}_Z^N(\mathbf{X}^N) \| P_Z\right)\right| < \sqrt{\frac{Nc_N^2}{2}}\sqrt{\log\left(\frac{2}{\delta}\right)}.$$

We will show that $c_N \leq kN^{-1/2}\psi(N)$ for $k$ and $\psi(N)$ depending on $f$. The statement of Theorem 3 is of this form. Note that in order to show that

$$\left| D_f\left(\hat{Q}_Z^N(\mathbf{X}^N) \| P_Z\right) - D_f\left(\hat{Q}_Z^N(\mathbf{X}^{N'}) \| P_Z\right)\right| \leq c_N, \tag{7}$$

it is sufficient to prove that

$$D_f\left(\hat{Q}_Z^N(\mathbf{X}^N) \| P_Z\right) - D_f\left(\hat{Q}_Z^N(\mathbf{X}^{N'}) \| P_Z\right) \leq c_N \tag{8}$$

since the symmetry in $\mathbf{X}^N \leftrightarrow \mathbf{X}^{N'}$ implies that

$$-D_f\left(\hat{Q}_Z^N(\mathbf{X}^N) \| P_Z\right) + D_f\left(\hat{Q}_Z^N(\mathbf{X}^{N'}) \| P_Z\right) \leq c_N \tag{9}$$

and thus implies Inequality 7. The remainder of this proof is therefore devoted to showing that Inequality 8 holds for each divergence.

We will make use of the fact that $\chi^2\left(Q_{Z|x} \| P_Z\right) \leq C \implies \left\|\frac{q(z|x)}{p(z)}\right\|_{L_2(P_Z)} \leq C + 1$

**The case that $D_f$ is the $\chi^2$-divergence, Total Variation or $D_{f_\beta}$ with $\beta > 1$:**

$$D_f\left(\hat{Q}_Z^N(\mathbf{X}^N)\|P_Z\right) - D_f\left(\hat{Q}_Z^N(\mathbf{X}^{N'})\|P_Z\right)$$

$$= \int f_0\left(\frac{d\hat{Q}_Z^N(\mathbf{X}^N)}{dP_Z}(z)\right) - f_0\left(\frac{d\hat{Q}_Z^N(\mathbf{X}^{N'})}{dP_Z}(z)\right)dP_Z(z)$$

$$\leq \int \left(\frac{\hat{q}_N(z;\mathbf{X}^N) - \hat{q}_N(z;\mathbf{X}^{N'})}{p(z)}\right) f_0'\left(\frac{\hat{q}_N(z;\mathbf{X}^N)}{p(z)}\right)dP_Z(z)$$

$$\leq \left\|\frac{\hat{q}_N(z;\mathbf{X}^N) - \hat{q}_N(z;\mathbf{X}^{N'})}{p(z)}\right\|_{L_2(P_Z)} \times \left\|f_0'\left(\frac{\hat{q}_N(z;\mathbf{X}^N)}{p(z)}\right)\right\|_{L_2(P_Z)} \qquad \text{(Cauchy-Schwartz)}$$

$$= \left\|\frac{1}{N}\frac{q(z|X_1) - q(z|X_1')}{p(z)}\right\|_{L_2(P_Z)} \times \left\|f_0'\left(\frac{\hat{q}_N(z;\mathbf{X}^N)}{p(z)}\right)\right\|_{L_2(P_Z)}$$

$$\leq \frac{1}{N}\left(\left\|\frac{q(z|X_1)}{p(z)}\right\|_{L_2(P_Z)} + \left\|\frac{q(z|X_1')}{p(z)}\right\|_{L_2(P_Z)}\right) \times \left\|f_0'\left(\frac{\hat{q}_N(z;\mathbf{X}^N)}{p(z)}\right)\right\|_{L_2(P_Z)}$$

$$\leq \frac{2(C+1)}{N}\left\|f_0'\left(\frac{\hat{q}_N(z;\mathbf{X}^N)}{p(z)}\right)\right\|_{L_2(P_Z)}.$$

By similar arguments as made in the proof of Theorem 2 considering the term $(ii)$, $\left\|f_0'\left(\frac{\hat{q}_N(z;\mathbf{X}^N)}{p(z)}\right)\right\|_{L_2(P_Z)} = \sqrt{\mathbb{E}_Z f_0'^2\left(\frac{\hat{q}_N(z;\mathbf{X}^N)}{p(z)}\right)} = O(1)$ thus we have the difference is upper-bounded by $c_N = \frac{k}{N}$ for some constant $k$. The only modification needed to the proof in Theorem 2 is the omission of all occurrences of $\mathbb{E}_{\mathbf{X}^N}$.

This holds for any $N > 0$.

**All other divergences.** Similar to the proof of Theorem 2, we write the difference as the sum of integrals over different mutually exclusive events that partition $\mathcal{Z}$. Denoting by $\gamma_N$ and $\delta_N$ scalars depending on $N$, we have that

$$D_f\left(\hat{Q}_Z^N(\mathbf{X}^N)\|P_Z\right) - D_f\left(\hat{Q}_Z^N(\mathbf{X}^{N'})\|P_Z\right)$$

$$= \int f_0\left(\frac{d\hat{Q}_Z^N(\mathbf{X}^N)}{dP_Z}(z)\right) - f_0\left(\frac{d\hat{Q}_Z^N(\mathbf{X}^{N'})}{dP_Z}(z)\right)dP_Z(z)$$

$$= \int f_0\left(\frac{d\hat{Q}_Z^N(\mathbf{X}^N)}{dP_Z}(z)\right) - f_0\left(\frac{d\hat{Q}_Z^N(\mathbf{X}^{N'})}{dP_Z}(z)\right)\mathbb{1}_{\left\{\frac{d\hat{Q}_Z^N(\mathbf{X}^N)}{dP_Z}(z)\leq\delta_N \text{ and } \frac{d\hat{Q}_Z^N(\mathbf{X}^{N'})}{dP_Z}(z)\leq\gamma_N\right\}}dP_Z(z) \qquad \text{(A)}$$

$$+ \int f_0\left(\frac{d\hat{Q}_Z^N(\mathbf{X}^N)}{dP_Z}(z)\right) - f_0\left(\frac{d\hat{Q}_Z^N(\mathbf{X}^{N'})}{dP_Z}(z)\right)\mathbb{1}_{\left\{\frac{d\hat{Q}_Z^N(\mathbf{X}^N)}{dP_Z}(z)\leq\delta_N \text{ and } \frac{d\hat{Q}_Z^N(\mathbf{X}^{N'})}{dP_Z}(z)>\gamma_N\right\}}dP_Z(z) \qquad \text{(B)}$$

$$+ \int f_0\left(\frac{d\hat{Q}_Z^N(\mathbf{X}^N)}{dP_Z}(z)\right) - f_0\left(\frac{d\hat{Q}_Z^N(\mathbf{X}^{N'})}{dP_Z}(z)\right)\mathbb{1}_{\left\{\frac{d\hat{Q}_Z^N(\mathbf{X}^N)}{dP_Z}(z)>\delta_N\right\}}dP_Z(z). \qquad \text{(C)}$$

We will consider each of the terms (A), (B) and (C) separately.

Later on, we will pick $\gamma_N$ and $\delta_N$ to be decreasing in $N$ such that $\delta_N < \gamma_N$. We will require $N$ sufficiently large so that $\gamma_N < 1$, so in the rest of this proof we will assume this to be the case and later on provide lower bounds on how large $N$ must be to ensure this.

(A): Recall that $f_0(x)$ is decreasing on the interval $[0, 1]$. Since $\gamma_N, \delta_N \leq 1$, the integrand is at most $f_0(0) - f_0(\gamma_N)$, and so

$$\text{(A)} \leq f_0(0) - f_0(\gamma_N)$$

$\boxed{\text{B}}$: Since $\delta_N \leq 1$, the integrand is at most $f_0(0)$ and so

$$\boxed{\text{B}} \leq f_0(0) \times \underbrace{\mathbb{P}_Z \left\{ \frac{d\hat{Q}_Z^N(\mathbf{X}^N)}{dP_Z}(z) \leq \delta_N \text{ and } \frac{d\hat{Q}_Z^N(\mathbf{X}^{N'})}{dP_Z}(z) > \gamma_N \right\}}_{\circledast}$$

We will bound $\mathbb{P}_Z \circledast = 0$ using Chebyshev's inequality. Noting that

$$\frac{\hat{q}_N(z; \mathbf{X}^N)}{p(z)} = \frac{\hat{q}_N(z; \mathbf{X}^{N'})}{p(z)} - \frac{1}{N}\frac{q(z|X_1')}{p(z)} + \frac{1}{N}\frac{q(z|X_1)}{p(z)},$$

and using the fact that $\frac{q(z|X_1)}{p(z)} > 0$ it follows that

$$\circledast \implies \gamma_N - \frac{1}{N}\frac{q(z|X_1')}{p(z)} + \frac{1}{N}\frac{q(z|X_1)}{p(z)} < \delta_N$$

$$\iff (\gamma_N - \delta_N)N + \frac{q(z|X_1)}{p(z)} < \frac{q(z|X_1')}{p(z)}$$

$$\implies (\gamma_N - \delta_N)N < \frac{q(z|X_1')}{p(z)}$$

$$\implies (\gamma_N - \delta_N)N - 1 < \frac{q(z|X_1')}{p(z)} - 1.$$

where the penultimate line follows from the fact that $q(z|X_1)/p(z) \geq 0$. It follows that

$$\mathbb{P}_Z \circledast \leq \mathbb{P}_Z \left\{ \frac{q(z|X_1')}{p(z)} - 1 > (\gamma_N - \delta_N)N - 1 \right\}$$

$$\leq \mathbb{P}_Z \left\{ \left| \frac{q(z|X_1')}{p(z)} - 1 \right| > (\gamma_N - \delta_N)N - 1 \right\}.$$

Denote by $\sigma^2(X) = \mathrm{Var}_Z \left[ \frac{q(z|X)}{p(z)} \right] = \mathbb{E}_Z \frac{q^2(z|X)}{p^2(z)} - 1 \leq C$. We have by Chebyshev that for any $t > 0$,

$$\mathbb{P}_Z \left\{ \left| \frac{q(z|X)}{p(z)} - 1 \right| > t \right\} \leq \frac{\sigma^2(X)}{t^2}$$

and so setting $t = (\gamma_N - \delta_N)N - 1$ yields

$$\mathbb{P}_Z \circledast \leq \frac{\sigma^2(X)}{((\gamma_N - \delta_N)N - 1)^2} \leq \frac{C}{((\gamma_N - \delta_N)N - 1)^2}$$

It follow that

$$\boxed{\text{B}} \leq f_0(0) \frac{C}{((\gamma_N - \delta_N)N - 1)^2}$$

$\boxed{\text{C}}$: Similar to the proof of Theorem 2, we can upper bound this term by the product of two terms, one of which is independent of the choice of divergence. The other term will be treated separately for each divergence considered.

$$\text{C} = \int f_0\left(\frac{\hat{q}_N(z;\mathbf{X}^N)}{p(z)}\right) - f_0\left(\frac{\hat{q}_N(z;\mathbf{X}^{N'})}{p(z)}\right) \mathbb{1}_{\left\{\frac{\hat{q}_N(z;\mathbf{X}^N)}{p(z)}>\delta_N\right\}} dP_Z(z)$$

$$\leq \int \left(\frac{\hat{q}_N(z;\mathbf{X}^N)}{p(z)} - \frac{\hat{q}_N(z;\mathbf{X}^{N'})}{p(z)}\right) f_0'\left(\frac{\hat{q}_N(z;\mathbf{X}^N)}{p(z)}\right) \mathbb{1}_{\left\{\frac{\hat{q}_N(z;\mathbf{X}^N)}{p(z)}>\delta_N\right\}} dP_Z(z) \quad \text{(Convexity of } f_0\text{)}$$

$$= \int \frac{1}{N}\frac{q(z|X_1) - q(z|X_1')}{p(z)} f_0'\left(\frac{\hat{q}_N(z;\mathbf{X}^N)}{p(z)}\right) \mathbb{1}_{\left\{\frac{\hat{q}_N(z;\mathbf{X}^N)}{p(z)}>\delta_N\right\}} dP_Z(z)$$

$$\leq \left\|\frac{1}{N}\frac{q(z|X_1) - q(z|X_1')}{p(z)}\right\|_{L_2(P_Z)} \left\|f_0'\left(\frac{\hat{q}_N(z;\mathbf{X}^N)}{p(z)}\right) \mathbb{1}_{\left\{\frac{\hat{q}_N(z;\mathbf{X}^N)}{p(z)}>\delta_N\right\}}\right\|_{L_2(P_Z)} \quad \text{(Cauchy-Schwartz)}$$

$$\leq \frac{2(C+1)}{N} \underbrace{\sqrt{\int f_0'^2\left(\frac{\hat{q}_N(z;\mathbf{X}^N)}{p(z)}\right) \mathbb{1}_{\left\{\frac{\hat{q}_N(z;\mathbf{X}^N)}{p(z)}>\delta_N\right\}} p(z)dz}}_{\text{(*)}} \quad \text{(Boundedness of } \left\|\frac{q(z|x)}{p(z)}\right\|_{L_2(P_Z)}\text{)}$$

The term (*) will be treated separately for each divergence.

**KL:** By Lemma 2, there exists a function $h_{\delta_N}(x)$ that is positive and concave on $[0,\infty)$ and is an upper bound of $f_0'^2(x)$ on $[\delta_N, \infty)$ with $h_{\delta_N}(1) = \log^2(\delta_N) + \frac{2}{e}$.

$$\text{(*)}^2 \leq \int h_{\delta_N}\left(\frac{\hat{q}_N(z;\mathbf{X}^N)}{p(z)}\right) \mathbb{1}_{\left\{\frac{\hat{q}_N(z;\mathbf{X}^N)}{p(z)}>\delta_N\right\}} p(z)dz \quad (h_{\delta_N} \text{ upper bounds } f'^2 \text{ on } (\delta_N, \infty))$$

$$\leq \int h_{\delta_N}\left(\frac{\hat{q}_N(z;\mathbf{X}^N)}{p(z)}\right) p(z)dz \quad (h_{\delta_N} \text{ non-negative on } [0,\infty))$$

$$\leq h_{\delta_N}\left(\int \frac{\hat{q}_N(z;\mathbf{X}^N)}{p(z)} p(z)dz\right) \quad (h_{\delta_N} \text{ concave})$$

$$= h_{\delta_N}(1)$$

$$= \log^2(\delta_N) + \frac{2}{e}$$

$$\implies \text{C} \leq \frac{2(C+1)}{N}\sqrt{\log^2(\delta_N) + \frac{2}{e}}.$$

Putting together the separate integrals and setting $\delta_N = \frac{1}{N^{2/3}}$ and $\gamma_N = \frac{2}{N^{2/3}}$, we have that

$$D_f\left(\hat{Q}_Z^N(\mathbf{X}^N)\|P_Z\right) - D_f\left(\hat{Q}_Z^N(\mathbf{X}^{N'})\|P_Z\right)$$

$$= \text{A} + \text{B} + \text{C}$$

$$\leq f_0(0) - f_0(\gamma_N) + \frac{f_0(0)C}{((\gamma_N - \delta_N)N - 1)^2} + \frac{2(C+1)}{N}\sqrt{\log^2(\delta_N) + \frac{2}{e}}$$

$$= \gamma_N - \gamma_N \log \gamma_N + \frac{f_0(0)C}{((\gamma_N - \delta_N)N - 1)^2} + \frac{2(C+1)}{N}\sqrt{\log^2(\delta_N) + \frac{2}{e}}$$

$$= \frac{2}{N^{2/3}} - \frac{2}{N^{2/3}}\log\left(\frac{2}{N^{2/3}}\right) + \frac{f_0(0)C}{(N^{1/3}-1)^2} + \frac{2(C+1)}{N}\sqrt{\frac{4}{9}\log^2(N) + \frac{2}{e}}$$

$$\leq \frac{2}{N^{2/3}} - \frac{2}{N^{2/3}}\log\left(\frac{2}{N^{2/3}}\right) + \frac{9f_0(0)C}{4N^{2/3}} + \frac{2(C+1)}{N}\sqrt{\frac{4}{9}\log^2(N) + \frac{2}{e}}$$

$$= \frac{k_1}{N^{2/3}} + \frac{k_2 \log N}{N^{2/3}} + \frac{k_3\sqrt{\log^2 N + \frac{9}{2e}}}{N}$$

$$\leq (k_1 + k_2 + 2k_3)\frac{\log N}{N^{2/3}}$$

where $k_1, k_2$ and $k_3$ are constants depending on $C$. The second inequality holds if $N^{1/3} - 1 > \frac{N^{1/3}}{3} \iff N > \left(\frac{3}{2}\right)^3 < 4$ and the third inequality holds if $N \geq 4$

The assumption that $\delta_N, \gamma_N \leq 1$ holds if $N > 2^{3/2}$ and so holds if $N \geq 3$.

This leads to $Nc_N^2 = \frac{\log^2 N}{N^{1/3}}$ for $N > 3$.

**Squared Hellinger.** In this case similar reasoning to the other divergences leads to a bound that is worse than $O\left(\frac{1}{\sqrt{N}}\right)$ and thus $Nc_N^2$ is bigger than $O(1)$ leading to a trivial concentration result.

$\alpha$**-divergence with** $\alpha \in \left(\frac{1}{3}, 1\right)$**.** Following similar reasoning to the proof of Theorem 2 for the $\alpha$-divergence case, we use the function $h_{\delta_N}(x)$ provided by Lemma 4 to derive the following upper bound:

$$\text{\textcircled{C}} \leq \frac{2(C+1)}{N} \cdot \frac{2\sqrt{1 + (C+1)^2}\left(\delta_N^{\frac{\alpha-1}{2}} - 1\right)}{(\alpha - 1)(\delta_N - 1)}.$$

Setting $\delta_N = \frac{1}{N^{\frac{4}{\alpha+5}}}$ and $\gamma_N = \frac{2}{N^{\frac{4}{\alpha+5}}}$,

$$D_f\left(\hat{Q}_Z^N(\mathbf{X}^N)\|P_Z\right) - D_f\left(\hat{Q}_Z^N(\mathbf{X}^{N'})\|P_Z\right)$$
$$= \text{\textcircled{A}} + \text{\textcircled{B}} + \text{\textcircled{C}}$$

$$\leq f_0(0) - f_0(\gamma_N) + \frac{f_0(0)C}{((\gamma_N - \delta_N)N - 1)^2} + \frac{2(C+1)}{N}\frac{2\sqrt{1+(C+1)^2}\left(\delta_N^{\frac{\alpha-1}{2}} - 1\right)}{(1-\alpha)(1-\delta_N)}$$

$$\leq f_0(0) - f_0(\gamma_N) + \frac{t^2 f_0(0)C}{(t-1)^2(\gamma_N - \delta_N)^2 N^2} + \frac{2(C+1)}{N}\frac{2\sqrt{1+(C+1)^2}\left(\delta_N^{\frac{\alpha-1}{2}} - 1\right)}{(1-\alpha)(1-\delta_N)}$$

$$\leq f_0(0) - f_0(\gamma_N) + \frac{t^2 f_0(0)C}{(t-1)^2(\gamma_N - \delta_N)^2 N^2} + \frac{2(C+1)}{N}\frac{4\sqrt{1+(C+1)^2}\delta_N^{\frac{\alpha-1}{2}}}{(1-\alpha)}$$

$$\leq k_1 \gamma_N^{\frac{\alpha+1}{2}} + k_2 \gamma_N + \frac{k_3}{(\gamma_N - \delta_N)^2 N^2} + \frac{k_4 \delta_N^{\frac{\alpha-1}{2}}}{N}$$

$$= \frac{k_1}{N^{\frac{2\alpha+2}{\alpha+5}}} + \frac{k_2}{N^{\frac{4}{\alpha+5}}} + \frac{k_3}{N^{\frac{2\alpha-2}{\alpha+5}}} + \frac{k_4}{N^{\frac{3\alpha+3}{\alpha+5}}}$$

$$\leq \frac{k_1 + k_2 + k_3 + k_4}{N^{\frac{2\alpha+2}{\alpha+5}}}$$

where $t$ is any positive number and where the second inequality holds if $N^{\frac{2\alpha+2}{\alpha+5}} - 1 > \frac{N^{\frac{2\alpha+2}{\alpha+5}}}{t} \iff N > \left(\frac{t}{t-1}\right)^{\frac{\alpha+5}{2\alpha+21}}$. For $\alpha \in \left(\frac{1}{3}, 1\right)$ we have $\frac{\alpha+5}{2\alpha+2} \in \left(\frac{3}{2}, 2\right)$. If we take $t = 100$ then $N > 1$ suffices for any $\alpha$.

The third inequality holds if $1 - \delta_N > \frac{1}{2} \iff N > 2^{\frac{\alpha+5}{4}}$ and so holds if $N > 3$.

The assumption that $\delta_N, \gamma_N \leq 1$ holds if $N > 4^{\frac{\alpha+5}{4}} \leq 8$ and so holds if $N > 8$.

Thus, this leads to $Nc_N^2 = \frac{k}{N^{\frac{3\alpha-1}{\alpha+5}}}$ for $N > 8$.

**Jensen-Shannon.** Following similar reasoning to the proof of Theorem 2 for the $\alpha$-divergence case, we use the function $h_{\delta_N}(x)$ provided by Lemma 5 to derive the following upper bound:

$$\text{\textcircled{C}} \leq \frac{2(C+1)}{N} \cdot \sqrt{5}\log\left(\frac{1}{\delta_N}\right).$$

Setting $\delta_N = \frac{1}{N^{2/3}}$ and $\gamma_N = \frac{2}{N^{2/3}}$,

$$D_f\left(\hat{Q}_Z^N(\mathbf{X}^N)\|P_Z\right) - D_f\left(\hat{Q}_Z^N(\mathbf{X}^{N'})\|P_Z\right)$$

$$= \text{Ⓐ} + \text{Ⓑ} + \text{Ⓒ}$$

$$\leq f_0(0) - f_0(\gamma_N) + \frac{f_0(0)C}{((\gamma_N - \delta_N)N - 1)^2} + \frac{2(C+1)}{N} \cdot \log\left(\frac{1}{\delta_N}\right)$$

$$\leq \gamma_N \log\left(\frac{1+\gamma_N}{2\gamma_N}\right) + \log(1+\gamma_N) + \frac{f_0(0)C}{((\gamma_N - \delta_N)N - 1)^2} + \frac{2(C+1)}{N} \cdot \log\left(\frac{1}{\delta_N}\right).$$

Using the fact that $\log(1+\gamma_N) \leq \gamma_N$, we obtain the following upper bound:

$$\leq \gamma_N^2 + \gamma_N(1 - \log 2) - \gamma_N \log \gamma_N + \frac{f_0(0)C}{((\gamma_N - \delta_N)N - 1)^2} + \frac{2(C+1)}{N} \cdot \log\left(\frac{1}{\delta_N}\right)$$

$$= \frac{k_1}{N^{4/3}} + \frac{k_2}{N^{2/3}} + \frac{k_3 \log N}{N^{2/3}} + \frac{k_4}{(N^{1/3} - 1)^2} + \frac{k_5 \log N}{N^{2/3}}$$

$$= \frac{k_1}{N^{4/3}} + \frac{k_2}{N^{2/3}} + \frac{k_3 \log N}{N^{2/3}} + \frac{k_4}{(N^{1/3} - 1)^2} + \frac{k_5 \log N}{N^{2/3}}$$

$$\leq \frac{k_1}{N^{4/3}} + \frac{k_2}{N^{2/3}} + \frac{k_3 \log N}{N^{2/3}} + \frac{100k_4}{81N^{2/3}} + \frac{k_5 \log N}{N^{2/3}}$$

$$\leq (k_1 + k_2 + k_3 + k_4' + k_5)\frac{\log N}{N^{2/3}}$$

where the penultimate inequality holds if $N^{1/3} - 1 > \frac{N^{1/3}}{10} \iff N > \left(\frac{10}{9}\right)^3$ which is satisfied if $N > 1$ and the last inequality is true if $N > 1$.

The assumption that $\delta_N, \gamma_N \leq 1$ holds if $N > 2^{3/2}$ and so holds if $N \geq 3$.

This leads to $Nc_N^2 = \frac{\log^2 N}{N^{1/3}}$ for $N > 2$.

$f_\beta$**-divergence, $\beta \in \left(\frac{1}{2}, 1\right)$.** Following similar reasoning to the proof of Theorem 2 for the $\alpha$-divergence case, we use the function $h_{\delta_N}(x)$ provided by Lemma 6 to derive the following upper bound:

$$\text{Ⓒ} \leq \frac{2(C+1)}{N} \cdot \frac{\beta}{1-\beta} \cdot 2^{\frac{2+\beta}{2\beta}} \delta_N^{\beta-1}.$$

Setting $\delta_N = \frac{1}{N^{2/3}}$ and $\gamma_N = \frac{2}{N^{2/3}}$,

$$D_f\left(\hat{Q}_Z^N(\mathbf{X}^N)\|P_Z\right) - D_f\left(\hat{Q}_Z^N(\mathbf{X}^{N'})\|P_Z\right)$$

$$= \text{Ⓐ} + \text{Ⓑ} + \text{Ⓒ}$$

$$\leq f_0(0) - f_0(\gamma_N) + \frac{f_0(0)C}{((\gamma_N - \delta_N)N - 1)^2} + \frac{\beta}{1-\beta} \cdot 2^{\frac{2+\beta}{2\beta}} \delta_N^{\beta-1}$$

$$\leq \frac{\beta}{\beta-1} 2^{\frac{1-\beta}{\beta}} \gamma_N + \frac{f_0(0)C}{((\gamma_N - \delta_N)N - 1)^2} + \frac{\beta}{1-\beta} \cdot 2^{\frac{2+\beta}{2\beta}} \frac{\delta_N^{\beta-1}}{N}$$

$$= \frac{k_1}{N^{2/3}} + \frac{k_2}{(N^{1/3} - 1)^2} + \frac{k_3}{N^{\frac{2\beta+1}{3}}}$$

$$\leq \frac{k_1}{N^{2/3}} + \frac{100k_2}{81N^{2/3}} + \frac{k_3}{N^{\frac{2\beta+1}{3}}}$$

$$\leq \frac{k_1 + k_2' + k_3}{N^{2/3}}$$

where the penultimate inequality holds if $N^{1/3} - 1 > \frac{N^{1/3}}{10} \iff N > \left(\frac{10}{9}\right)^3$ which is satisfied if $N > 1$.

The assumption that $\delta_N, \gamma_N \leq 1$ holds if $N > 2^{3/2}$ and so holds if $N \geq 3$.

This leads to $Nc_N^2 = \frac{1}{N^{1/3}}$ for $N > 2$.

$\square$

## B.6 Full statement and proof of Theorem 4

The statement of Theorem 4 in the main text was simplified for brevity. Below is the full statement, followed by its proof.

**Theorem 4.** *For any $\pi$,*

$$\mathbb{E}_{\mathbf{Z}^M,\mathbf{X}^N}[\hat{D}_f^M(\hat{Q}_Z^N\|P_Z)] = \mathbb{E}_{\mathbf{X}^N}\left[D_f\left(\hat{Q}_Z^N\|P_Z\right)\right].$$

*If either of the following conditions are satisfied:*

$(i)\ \pi(z|\mathbf{X}^N) = p(z), \qquad \mathbb{E}_X\left\|f\left(\frac{q(z|X)}{p(z)}\right)\right\|_{L_2(P_Z)}^2 < \infty, \qquad \mathbb{E}_X\left\|\frac{q(z|X)}{p(z)}\right\|_{L_2(P_Z)}^2 < \infty$

$(ii)\ \pi(z|\mathbf{X}^N) = \hat{q}_N(z), \qquad \mathbb{E}_X\left\|f\left(\frac{q(z|X)}{p(z)}\right)\frac{p(z)}{q(z|X)}\right\|_{L_2(Q_{Z|X})}^2 < \infty, \quad \mathbb{E}_X\left\|\frac{p(z)}{q(z|X)}\right\|_{L_2(Q_{Z|X})}^2 < \infty$

*then, denoting by $\psi(N)$ the rate given in Table 2, we have*

$$Var_{\mathbf{Z}^M,\mathbf{X}^N}\left[\hat{D}_f^M(\hat{Q}_Z^N\|P_Z)\right] = O\left(M^{-1}\right) + O\left(\psi(N)^2\right)$$

In proving Theorem 4 we will make use of the following lemma.

**Lemma 7.** *For any $f_0(x)$, the functions $f_0(x)^2$ and $\frac{f_0(x)^2}{x}$ are convex on $(0,\infty)$.*

*Proof.* To see that $f_0(x)^2$ is convex, observe that

$$\frac{d^2}{dx^2}f_0(x)^2 = 2\left(f_0(x)f_0''(x) + f_0'(x)^2\right)$$

All of these terms are postive for $x > 0$. Indeed, since $f_0(x)$ is convex for $x > 0$, $f_0''(x) \geq 0$. By construction of $f_0$, $f_0(x) \geq 0$ for $x > 0$. Thus $f_0(x)^2$ has non-negative derivative and is thus convex on $(0,\infty)$.

To see that $\frac{f_0(x)^2}{x}$ is convex, observe that

$$\frac{d^2}{dx^2}\frac{f_0(x)^2}{x} = \frac{2}{x}\left(f_0(x)f_0''(x) + \left(f_0'(x) - \frac{f_0(x)}{x}\right)^2\right).$$

By the same arguments above, this is positive for $x > 0$ and thus $\frac{f_0(x)^2}{x}$ is convex for $x > 0$. $\square$

*Proof.* (Theorem 4) For the expectation, observe that

$$\mathbb{E}_{\mathbf{Z}^M,\mathbf{X}^N}\hat{D}_f^M(\hat{Q}_Z^N\|P_Z) = \mathbb{E}_{\mathbf{X}^N}\left[\mathbb{E}_{\mathbf{Z}^M\overset{i.i.d.}{\sim}\pi(z|\mathbf{X}^N)}\hat{D}_f^M(\hat{Q}_Z^N\|P_Z)\right]$$

$$= \mathbb{E}_{\mathbf{X}^N}\left[\mathbb{E}_{z\sim\pi(z|\mathbf{X}^N)}f\left(\frac{\hat{q}_N(z)}{p(z)}\right)\frac{p(z)}{\pi(z|\mathbf{X}^N)}\right]$$

$$= \mathbb{E}_{\mathbf{X}^N}\left[D_f\left(\hat{Q}_Z^N\|P_Z\right)\right].$$

For the variance, by the law of total variance we have that

$$Var_{\mathbf{Z}^M,\mathbf{X}^N}\left[\hat{D}_f^M(\hat{Q}_Z^N\|P_Z)\right]$$

$$= \mathbb{E}_{\mathbf{X}^N}Var_{\mathbf{Z}^M\overset{i.i.d.}{\sim}\pi(z|\mathbf{X}^N)}\hat{D}_f^M(\hat{Q}_Z^N\|P_Z) + Var_{\mathbf{X}^N}\mathbb{E}_{\mathbf{Z}^M\overset{i.i.d.}{\sim}\pi(z|\mathbf{X}^N)}\hat{D}_f^M(\hat{Q}_Z^N\|P_Z)$$

$$= \frac{1}{M}\underbrace{\mathbb{E}_{\mathbf{X}^N}Var_{\pi(z|\mathbf{X}^N)}\left[f\left(\frac{\hat{q}_N(z)}{p(z)}\right)\frac{p(z)}{\pi(z|\mathbf{X}^N)}\right]}_{(i)} + \underbrace{Var_{\mathbf{X}^N}\left[D_f\left(\hat{Q}_Z^N\|P_Z\right)\right]}_{(ii)}.$$

Consider term $(ii)$. The concentration results of Theorem 3 imply bounds on $(ii)$, since for a random variable $X$,

$$\mathrm{Var}X = \mathbb{E}(X - EX)^2$$
$$= \int_0^\infty \mathbb{P}\left((X - \mathbb{E}X)^2 > t\right) dt$$
$$= \int_0^\infty \mathbb{P}\left(|X - \mathbb{E}X| > \sqrt{t}\right) dt.$$

It follows therefore that

$$\mathrm{Var}_{\mathbf{X}^N}\left[D_f\left(\hat{Q}_Z^N \| P_Z\right)\right] \leq \int_0^\infty 2\exp\left(-\frac{k}{\psi(N)^2}t\right) dt$$
$$= O\left(\psi(N)^2\right)$$

where $\psi(N)$ is given by Table 2.

Next we consider $(i)$ and show that it is bounded independent of $N$, and so the component of the variance due to this term is $O\left(\frac{1}{M}\right)$. In the case that $\pi(z|\mathbf{X}^N) = p(z)$,

$$(i) \leq \mathop{\mathbb{E}}_{\mathbf{X}^N} \mathop{\mathbb{E}}_{p(z)}\left[f\left(\frac{\hat{q}_N(z)}{p(z)}\right)^2\right]$$

$$= \mathop{\mathbb{E}}_{\mathbf{X}^N} \mathop{\mathbb{E}}_{p(z)}\left[\left(f_0\left(\frac{\hat{q}_N(z)}{p(z)}\right) + f'(1)\left(\frac{\hat{q}_N(z)}{p(z)} - 1\right)\right)^2\right]$$

$$\leq \mathop{\mathbb{E}}_{\mathbf{X}^N} \mathop{\mathbb{E}}_{p(z)}\left[f_0\left(\frac{\hat{q}_N(z)}{p(z)}\right)^2\right] + f'(1)^2 \mathop{\mathbb{E}}_{\mathbf{X}^N} \mathop{\mathbb{E}}_{p(z)}\left[\left(\frac{\hat{q}_N(z)}{p(z)} - 1\right)^2\right]$$

$$+ 2f'(1)\sqrt{\mathop{\mathbb{E}}_{\mathbf{X}^N} \mathop{\mathbb{E}}_{p(z)}\left[f_0\left(\frac{\hat{q}_N(z)}{p(z)}\right)^2\right]} \times \sqrt{\mathop{\mathbb{E}}_{\mathbf{X}^N} \mathop{\mathbb{E}}_{p(z)}\left[\left(\frac{\hat{q}_N(z)}{p(z)} - 1\right)^2\right]}$$

$$\leq \mathop{\mathbb{E}}_{X} \mathop{\mathbb{E}}_{p(z)}\left[f_0\left(\frac{q(z|X)}{p(z)}\right)^2\right] + f'(1)^2 \mathop{\mathbb{E}}_{X} \mathop{\mathbb{E}}_{p(z)}\left[\left(\frac{q(z|X)}{p(z)} - 1\right)^2\right]$$

$$+ 2f'(1)\sqrt{\mathop{\mathbb{E}}_{X} \mathop{\mathbb{E}}_{p(z)}\left[f_0\left(\frac{q(z|X)}{p(z)}\right)^2\right]} \times \sqrt{\mathop{\mathbb{E}}_{X} \mathop{\mathbb{E}}_{p(z)}\left[\left(\frac{q(z|X)}{p(z)} - 1\right)^2\right]}$$

The penultimate inequality follows by application of Cauchy-Schwartz. The last inequality follows by Proposition 1 applied to $D_{f_0^2}$ and $D_{(x-1)^2}$, using the fact that the functions $f_0^2(x)$ and $(x-1)^2$ are convex and are zero at $x = 1$ (see Lemma 7). By assumption, $\mathbb{E}_X \mathbb{E}_{p(z)}\left[\left(\frac{q(z|X)}{p(z)} - 1\right)^2\right] < \infty$. Consider the other term:

$$\mathop{\mathbb{E}}_{X} \mathop{\mathbb{E}}_{p(z)}\left[f_0\left(\frac{q(z|X)}{p(z)}\right)^2\right] = \mathop{\mathbb{E}}_{X} \mathop{\mathbb{E}}_{p(z)}\left[\left(f\left(\frac{q(z|X)}{p(z)}\right) - f'(1)\left(\frac{q(z|X)}{p(z)} - 1\right)\right)^2\right]$$

$$\leq \mathop{\mathbb{E}}_{X} \mathop{\mathbb{E}}_{p(z)}\left[f\left(\frac{q(z|X)}{p(z)}\right)^2\right] + f'(1)^2 \mathop{\mathbb{E}}_{X} \mathop{\mathbb{E}}_{p(z)}\left[\left(\frac{q(z|X)}{p(z)} - 1\right)^2\right]$$

$$+ 2f'(1)\sqrt{\mathop{\mathbb{E}}_{X} \mathop{\mathbb{E}}_{p(z)}\left[f\left(\frac{q(z|X)}{p(z)}\right)^2\right]} \times \sqrt{\mathop{\mathbb{E}}_{X} \mathop{\mathbb{E}}_{p(z)}\left[\left(\frac{q(z|X)}{p(z)} - 1\right)^2\right]}$$

$$< \infty$$

The inequality follows by Cauchy-Schwartz. All terms are finite by assumption. Thus $(i) \leq K < \infty$ for some $K$ independent of $N$.

Now consider the case that $\pi(z|\mathbf{X}^N) = \hat{q}_N(z)$. Then, following similar (but algebraically more tedious) reasoning to the previous case, it can be shown that

$$(i) \leq \mathop{\mathbb{E}}_{X} \mathop{\mathbb{E}}_{p(z)} \left[ f_0 \left( \frac{q(z|X)}{p(z)} \right)^2 \frac{p(z)}{q(z|X)} \right] + f'(1)^2 \mathop{\mathbb{E}}_{X} \mathop{\mathbb{E}}_{p(z)} \left[ \left( \sqrt{\frac{q(z|X)}{p(z)}} - \sqrt{\frac{p(z)}{q(z|X)}} \right)^2 \right]$$

$$+ 2f'(1) \sqrt{\mathop{\mathbb{E}}_{X} \mathop{\mathbb{E}}_{p(z)} \left[ f_0 \left( \frac{q(z|X)}{p(z)} \right)^2 \frac{p(z)}{q(z|X)} \right]} \times \sqrt{\mathop{\mathbb{E}}_{X} \mathop{\mathbb{E}}_{p(z)} \left[ \left( \sqrt{\frac{q(z|X)}{p(z)}} - \sqrt{\frac{p(z)}{q(z|X)}} \right)^2 \right]}$$

where Proposition 1 is applied to $D_{\frac{f_0^2(x)}{x}}$ and $D_{(\sqrt{x} - \frac{1}{\sqrt{x}})^2}$, using the fact that the functions $f_0^2(x)/x$ and $(\sqrt{x} - \frac{1}{\sqrt{x}})^2$ are convex and are zero at $x = 1$ (see Lemma 7). Noting that

$$\mathop{\mathbb{E}}_{X} \mathop{\mathbb{E}}_{p(z)} \left[ \left( \sqrt{\frac{q(z|X)}{p(z)}} - \sqrt{\frac{p(z)}{q(z|X)}} \right)^2 \right] = \mathop{\mathbb{E}}_{X} \mathop{\mathbb{E}}_{p(z)} \left[ \frac{q(z|X)}{p(z)} + \frac{p(z)}{q(z|X)} - 2 \right]$$

$$= \mathop{\mathbb{E}}_{X} \mathop{\mathbb{E}}_{p(z)} \left[ \frac{p(z)}{q(z|X)} - 1 \right] < \infty$$

where the inequality holds by assumption, it follows that

$$\mathop{\mathbb{E}}_{X} \mathop{\mathbb{E}}_{p(z)} \left[ f_0 \left( \frac{q(z|X)}{p(z)} \right)^2 \frac{p(z)}{q(z|X)} \right]$$

$$\leq \mathop{\mathbb{E}}_{X} \mathop{\mathbb{E}}_{p(z)} \left[ f \left( \frac{q(z|X)}{p(z)} \right)^2 \frac{p(z)}{q(z|X)} \right] + f'(1)^2 \mathop{\mathbb{E}}_{X} \mathop{\mathbb{E}}_{p(z)} \left[ \left( \sqrt{\frac{q(z|X)}{p(z)}} - \sqrt{\frac{p(z)}{q(z|X)}} \right)^2 \right]$$

$$+ 2f'(1) \sqrt{\mathop{\mathbb{E}}_{X} \mathop{\mathbb{E}}_{p(z)} \left[ f \left( \frac{q(z|X)}{p(z)} \right)^2 \frac{p(z)}{q(z|X)} \right]} \times \sqrt{\mathop{\mathbb{E}}_{X} \mathop{\mathbb{E}}_{p(z)} \left[ \left( \sqrt{\frac{q(z|X)}{p(z)}} - \sqrt{\frac{p(z)}{q(z|X)}} \right)^2 \right]}$$

$$< \infty.$$

where the first inequality holds by the definition of $f_0$ and Cauchy-Schwartz.

Thus $(i) \leq K < \infty$ for some $K$ independent of $N$ in both cases of $\pi$. $\qquad \square$

## B.7 Elaboration of Section 2.3: satisfaction of assumptions of theorems

Suppose that $P_Z$ is $\mathcal{N}(0, I_d)$ and $Q_{Z|X}$ is $\mathcal{N}(\mu(X), \Sigma(X))$ with $\Sigma$ diagonal. Suppose further that there exist constants $K, \epsilon > 0$ such that $\|\mu(X)\| \leq K$ and $\Sigma_{ii}(X) \in [\epsilon, 1]$ for all $i$.

By Lemma 8, it holds that $\chi^2(Q_{Z|x}, P_Z) < \infty$ for all $x \in \mathcal{X}$. By compactness of the sets in which $\mu(X)$ and $\Sigma(X)$ take value, it follows that there exists $C < \infty$ such that $\chi^2(Q_{Z|x}, P_Z) \leq C$ and thus the setting of Theorem 3 holds.

A similar argument based on compactness shows that the density ratio is uniformly bounded in $z$ and $x$: $q(z|x)/p(z) \leq C'$ for some $C' < \infty$. It therefore follows that the condition of Theorem 2 holds: $\int q^4(z|x)/p^4(z) dP(z) < C'^4 < \infty$.

We conjecture that the strong boundedness assumptions on $\mu(X)$ and $\Sigma(X)$ also imply the setting of Theorem 1 $\mathbb{E}_X [\chi^2(Q_{Z|X}, Q_Z)] < \infty$. Since the divergence $Q_Z$ explicitly depends on the data distribution, this is more difficult to verify than the conditions of Theorems 2 and 3.

The crude upper bound provided by convexity

$$\mathop{\mathbb{E}}_{X} [\chi^2(Q_{Z|X}, Q_Z)] \leq \mathop{\mathbb{E}}_{X} \mathop{\mathbb{E}}_{X'} [\chi^2(Q_{Z|X}, Q_{Z|X'})]$$

provides a sufficient (but very strong) set of assumptions under which it holds. Finiteness of the right hand side above would be implied, for instance, by demanding that $\|\mu(X)\| \leq K$ and $\Sigma_{ii}(X) \in [\frac{1}{2} + \epsilon, 1]$ for all $i$.

## C Empirical evaluation: further details

In this section with give further details about the synthetic and real-data experiments presented in Section 3.

## C.1 Synthetic experiments

### C.1.1 Analytical expressions for divergences between two Gaussians

The closed form expression for the $\chi^2$-divergence between two $d$-variate normal distributions can be found in Lemma 1 of [29]:

**Lemma 8.**

$$\chi^2\big(\mathcal{N}(\mu_1, \Sigma_1), \mathcal{N}(\mu_2, \Sigma_2)\big) = \frac{\det(\Sigma_1^{-1})}{\sqrt{\det(2\Sigma_1^{-1} - \Sigma_2^{-1})\det(\Sigma_2^{-1})}} \exp\left(\frac{1}{2}\mu_2'\Sigma_2^{-1}\mu_2 - \mu_1'\Sigma_1^{-1}\mu_1\right) \times$$

$$\times \exp\left(-\frac{1}{4}(2\mu_1'\Sigma_1^{-1} - \mu_2'\Sigma_2^{-1})\left(\frac{1}{2}\Sigma_2^{-1} - \Sigma_1^{-1}\right)^{-1}(2\Sigma_1^{-1}\mu_1 - \Sigma_2^{-1}\mu_2)\right) - 1.$$

As a corollary, the following also holds:

**Corollary 1.** *Chi square divergence between two d-variate Gaussian distributions both having covariance matrices proportional to identity can be computed as:*

$$\chi^2\big(\mathcal{N}(\mu, \sigma^2 I_d), \mathcal{N}(0, \beta^2 I_d)\big) = \left(\frac{\beta^2}{\sigma^2\sqrt{2\beta^2/\sigma^2 - 1}}\right)^d e^{\frac{\|\mu\|^2}{2\beta^2 - \sigma^2}} - 1$$

*assuming $2\beta^2 > \sigma^2$. Otherwise the divergence is infinite.*

The squared Hellinger divergence between two Gaussians is given in [33]:

**Lemma 9.**

$$H^2\big(\mathcal{N}(\mu_1, \Sigma_1), \mathcal{N}(\mu_2, \Sigma_2)\big)$$
$$= 1 - \frac{\det(\Sigma_1)^{1/4}\det(\Sigma_2)^{1/4}}{\det\left(\frac{\Sigma_1 + \Sigma_2}{2}\right)^{1/2}} \exp\left\{-\frac{1}{8}(\mu_1 - \mu_2)^T\left(\frac{\Sigma_1 + \Sigma_2}{2}\right)^{-1}(\mu_1 - \mu_2)\right\}.$$

The KL-divergence between two $d$-variate Gaussians is:

**Lemma 10.**

$$\mathrm{KL}\big(\mathcal{N}(\mu_1, \Sigma_1), \mathcal{N}(\mu_2, \Sigma_2)\big) = \frac{1}{2}\left(tr\left(\Sigma_2^{-1}\Sigma_1\right) + (\mu_2 - \mu_1)^\intercal\Sigma_2^{-1}(\mu_2 - \mu_1) - d + \log\frac{|\Sigma_2|}{|\Sigma_1|}\right).$$

### C.1.2 Further experimental details

We take $Q_{Z|X=x}^\lambda = \mathcal{N}\left(A_\lambda x + b_\lambda, \epsilon^2 I_d\right)$ and $P_X = \mathcal{N}\left(0, I_{20}\right)$. This results in $Q_Z^\lambda = \mathcal{N}\left(b_\lambda, A_\lambda A_\lambda^\intercal + \epsilon^2 I_d\right)$. We chose $\epsilon = 0.5$ and used $\lambda \in [-2, 2]$. $P_Z = \mathcal{N}(0, I_d)$.

$A_\lambda$ and $b_\lambda$ were determined as follows: Define $A_1$ to be the $(d, 20)$-dimensional matrix with 1's on the main diagonal, and let $A_0$ be similarly sized matrix with entries randomly sampled i.i.d. unit Gaussians which is then normalised to have unit Frobenius norm. Let $v$ be a vector randomly sampled from the $d$-dimensional unit sphere. We then set $A_\lambda = \frac{1}{2}A_1 + \lambda A_0$ and $b_\lambda = \lambda v$.

$A_0$ and $v$ are sampled once for each dimension $d \in \{1, 4, 16\}$, such that the within each column of Figure 1, the distributions used are the same.

## C.2 Real-data experiments

### C.2.1 Variational Autoencoders (VAEs) and Wasserstein Autoencoders (WAEs)

Autoencoders are a general class of models typically used to learn compressed representations of high-dimensional data. Given a *data-space* $\mathcal{X}$ and low-dimensional *latent space* $\mathcal{Z}$, the goal is to learn an *encoder* mapping $\mathcal{X} \to \mathcal{Z}$ and *generator* (or *decoder*[3]) mapping $\mathcal{Z} \to \mathcal{X}$. The objectives used to train these two components always involve some kind of reconstruction loss measuring how corrupted a datum becomes after mapping through both the encoder and generator, and often some kind of regularization.

Representing by $\theta$ and $\eta$ the parameters of the encoder and generator respectively, the objective functions of VAEs and WAEs are:

$$L^{\mathrm{VAE}}(\theta, \eta) = \mathop{\mathbb{E}}_{X} \left[ \mathop{\mathbb{E}}_{q_\theta(Z|X)} \log p_\eta(X|Z) + \mathrm{KL}\left( Q^\theta_{Z|X} \| P_Z \right) \right]$$

$$L^{\mathrm{WAE}}(\theta, \eta) = \mathop{\mathbb{E}}_{X} \mathop{\mathbb{E}}_{q_\theta(Z|X)} c(X, G_\eta(Z)) + \lambda \cdot D(Q^\theta_Z \| P_Z)$$

For VAEs, both encoder $Q^\theta_{Z|X}$ and generator $p_\eta$ are *stochastic* mappings taking an input and mapping it to a distribution over the output space. In WAEs, only the encoder $Q^\theta_{Z|X}$ is stochastic, while the generator $G_\eta$ is deterministic. $c$ is a cost function, $\lambda$ is a hyperparameter and $D$ is any divergence.

A common assumption made for VAEs is that the generator outputs a Gaussian distribution with fixed diagonal covariance and mean $\mu(z)$ that is a function of the input $z$. In this case, the $\log p_\eta(X|z)$ term can be written as the $l_2^2$ (i.e. square of the $l_2$ distance) between $X$ and its reconstruction after encoding and re-generating $\mu(z)$. If the cost function of the WAE is chosen to be $l_2^2$, then the left hand terms of the VAE and WAE losses are the same. That is, in this particular case, $L^{\mathrm{VAE}}$ and $L^{\mathrm{WAE}}$ differ only in their regularizers.

The penalty of the VAE was shown by [19] to be equivalent to $\mathrm{KL}(Q^\theta_Z \| P_Z) + I(X, Z)$ where $I(X, Z)$ is the mutual information of a sample and its encoding. For the WAE penalty, there is a choice of which $D(Q^\theta_Z \| P_Z)$ to use; it must only be possible to practically estimate it. In the experiments used in this paper, we considered models trained with the Maximum Mean Discrepency (MMD) [13], a kernel-based distance on distributions, and a divergence estimated using a GAN-style classifier [12] leading to WAE-MMD and WAE-GAN respectively, following [38].

### C.2.2 Further experimental details

We took a corpus of VAE, WAE-GAN and WAE-MMD models that had been trained with a large variety of hyperparameters including learning rate, latent dimension (32, 64, 128), architecture (ResNet/DCGAN), scalar factor for regulariser, and additional algorithm-specific hyperparameters: kernel bandwidth for WAE-MMD and learning rate of discriminator for WAE-GAN. In total, 60 models were trained of each type (WAE-MMD, WAE-GAN and VAE) leading to 180 models in total.

The small subset of six models exposed in Figures 2 and 3 were selected by a heuristic that we next describe. However, we note that qualitatively similar behaviour was found in all other models tested, and so the choice of models to display was somewhat arbitrary; we describe it nonetheless for completeness.

Recall that the objective functions of WAEs and VAEs both include a divergence between $Q^\theta_Z$ and $P_Z$. We were interested in considering models from the two extremes of the distribution matching: some models in which $Q^\theta_Z$ and $P_Z$ were close, some in which they were distant.

To determine whether $Q^\theta_Z$ and $P_Z$ in a model are close, we made use of FID [18] scores as a proxy that is independent of the particular divergences for training. The FID score between two distributions over images is obtained by pushing both distributions through to an intermediate feature layer of the *Inception* network. The resulting push-through distributions are approximated with Gaussians and the *Fréchet* distance between them is calculated. Denote by $G_\#(Q^\theta_Z)$ the distribution over reconstructed images, $G_\#(P_Z)$ the distribution over model samples and $Q_X$ the data distribution, where $G$ is the generator and $\#$ denotes the push-through operator. The quantity $\mathrm{FID}\left(Q_X, G_\#(Q^\theta_Z)\right)$ is a measure of quality (lower is better) of the reconstructed data, while $\mathrm{FID}\left(Q_X, G_\#(P_Z)\right)$ is a measure of quality of model samples.

The two FID scores being very different is an indication that $P_Z$ and $Q^\theta_Z$ are different. In contrast, if the two FID scores are similar, we cannot conclude that $P_Z$ and $Q^\theta_Z$ are the same, though it provides some evidence towards that fact. Therefore, in order to select a model in which matching between $P_Z$ and $Q^\theta_Z$ is poor, we pick one for which $\mathrm{FID}\left(Q_X, G_\#(Q^\theta_Z)\right)$ is small but $\mathrm{FID}\left(Q_X, G_\#(P_Z)\right)$ is large (good reconstructions; poor samples). In order to select a model in which matching between $P_Z$ and $Q^\theta_Z$ is good, we pick one for both FIDs are small (good reconstructions; good samples). We will refer to these settings as *poor matching* and *good matching* respectively.

Our goal was to pick models according to the following criteria. The six chosen should include: two from each model class (VAE, WAE-GAN, WAE-MMD), of which one from each should exhibit poor matching and one good matching; two from each dimension $d \in \{32, 64, 128\}$; three with the ResNet architecture and three with the DCGAN architecture. A set of models satisfying these criteria were selected by hand, but as noted previously we saw qualitatively similar results with the other models.

### C.2.3 Additional results for squared Hellinger distance

Figure 3 we display similar results to those displayed in Figure 2 of the main paper but with the $H^2$-divergence instead of the KL. An important point is that $H^2(A, B) \in [0, 2]$ for any probability distributions $A$ and $B$, and

Figure 3: Estimating $H^2(Q_Z^\theta \| P_Z)$ in pretrained autoencoder models with RAM-MC as a function of $N$ for $M = 10$ (**green**) and $M{=}1000$ (**red**) compared to ground truth (**blue**). Lines and error bars represent means and standard deviations over 50 trials. Plots depict $\log\left(2 - \hat{D}_{H^2}^M(\hat{Q}_Z^N \| P_Z)\right)$ since $H^2$ is close to 2 in all models. Omitted lower error bars correspond to error bars going to $-\infty$ introduced by $\log$. Note that the approximately *increasing* behaviour evident here corresponds to the expectation of RAM-MC *decreasing* as a function of $N$. Due to concavity of $\log$, the decrease in variance when increasing $M$ manifests itself as the **red** line ($M{=}1000$) being consistently above the **green** line ($M{=}10$).

due to considerations of scale we plot the estimated values $\log\left(2 - \hat{D}_{H^2}^M(\hat{Q}_Z^N \| P_Z)\right)$. Decreasing bias in $N$ of RAM-MC therefore manifests itself as the lines *increasing* in Figure 3. Concavity of $\log$ means that the reduction in variance when increasing $M$ results in RAM-MC with $M{=}1000$ being above RAM-MC with $M{=}10$. Similar to those presented in the main part of the paper, these results therefore also support the theoretical findings of our work.

We additionally attempted the same experiment using the $\chi^2$-divergence but encountered numerical issues. This can be understood as a consequence of the inequality $e^{\text{KL}(A,B)} - 1 \leq \chi^2(A,B)$ for any distributions $A$ and $B$. From Figure 2 we see that the KL-divergence reaches values higher than 1000 which makes the corresponding value of the $\chi^2$-divergence larger than can be represented using double-precision floats.

## Footnotes

[3]In the VAE literature, the encoder and generator are sometimes referred to as the *inference network* and *likelihood model* respectively.