[Reviews · NeurIPS 2019]

Reviewer 1



There are several rooms for improvement of the proposed method. For practical use of the RAM-MC estimator, \hat{q}_N(z) should be known, however, this assumption seems restrictive. Several sampling methods work without known density functions, hence it is necessary to develop an estimator without knowing density functions. Also, the theoretical analysis may be improved. Since a main source of the error is from estimating density functions from both theoretical and practical aspects, hence an error of the estimator should be investigated without known density functions.

Reviewer 2



This paper discusses mainly theoretical convergence results for MC-based variational estimators of f-divergences. To the best of my knowledge, the presented theoretical convergence analysis of the f-divergence estimators is novel work. The results give nice and comprehensive rates of convergence of both the estimator's expectation, as well as its MC-estimator, for different f-functions. This convergence result gives theoretical justification to several well-established methods that can be interpreted as special cases of this work. The paper is well structured, very accessible and provides both high-level proof sketches and exhaustive proofs in the supplementary material. The work is in general well motivated and provides practical context to the theoretical results. As a minor drawback, the motivation and context seems to focus quite strongly on variational-inference related methods, while estimators of f-divergence are of interest in a much broader field as well. Limitations of the proposed results were recognized (such as constants still potentially depending on the dimensionality, but not in N^{-d}).

Reviewer 3



The paper considers an important problem of estimating f-divergences under a scenario which has a lot of potential applications. Pros: - The proposed estimator is simple to understand and implement. - The theoretical analysis is complete and many cases are examined. - Simulation considers many cases and contains real data and simulated data, as well as comparison of other methods. Cons - Some notations in the paper are confusing. (e.g. line 59, q() is multi-defined. Please use proper footnote to distinguish them) - N=1 in the simulation seems not very necessary. What is the purpose here? The other methods are all under N=500 - If there is an extra table comparing the proposed method with the convergence rate of the existing method that would be very helpful. Overall a good contribution to the venue and I would recommend the paper to be included. Edit: Thank the authors for the rebuttal. I appreciate the specific replies. I will keep my score as accept.

[Author Response · NeurIPS 2019]

We thank the reviewers for their time, and are happy with the generally positive view of our work.

Reviewers 2+3 find that our proposed $f$-divergence estimator is "applicable to many machine learning problems"
(R3) and gives "theoretical justification to several well-established methods" (R2). The "theoretical analysis [of our
estimator] is complete" (R3) and provides "nice and comprehensive rates of convergence of both the estimator's
expectation, as well as its MC-estimator, for different $f$-functions" (R2). Both reviewers recognise the significance of
our proposal and are in favour of acceptance.

It seems, however, that there is some disagreement between Reviewers 2+3 and Reviewer 1.

The disagreement seems to stem from Reviewer 1 having misunderstood a fundamental premise of the paper: namely,
that we study $f$-divergence estimation under strong assumptions (lines 55-64) which are nonetheless realistic in many
modern applications (lines 76-79, Section 4). Hence, although our setting "seems restrictive" (R1) it is in fact still very
much applicable. Moreover, it is precisely because of our assumptions that we derive superior rates compared to the
weak-assumption setting (lines 72-75). In light of this, Reviewer 1's requests to "provide additional theoretical analysis
for the estimator" and that "the estimator should be investigated without known density functions" for the conditionals
$Q_{Z|X}$ do not make much sense, simply because our proposed estimator can not be computed in this setting.

We hope that this clears up the misunderstanding and that Reviewer 1 will consider raising their rating.

Comments specifically for Reviewer 3:

• Thank you for spotting the overloaded notation. We will fix this to avoid ambiguity.

• We included RAM-MC with both $N = 1$ and $N = 500$ to show that increasing $N$ results in decreased bias
and variance in order to validate Theorems 1, 2 & 4. We discuss this in lines 216-217, but will update the
paragraph beginning line 192 to also explain why we do this. We did not include $N = 1$ for the other methods
because the plots would have become too cluttered.

• Table with other rates: this is a great suggestion, and we will definitely include it. Below is a table with rates
and assumptions beneath. We will update Table 1 in our paper accordingly.

Table 1: Rate of bias for estimators of $D_f(P, Q)$.

| $f$-divergence | KL | TV | $\chi^2$ | H$^2$ | JS | $D_{f_\beta}$ $\frac{1}{2}<\beta<1$ | $1<\beta<\infty$ | $D_{f_\alpha}$ $-1<\alpha<1$ |
|---|---|---|---|---|---|---|---|---|
| Krishnamurthy et al. [22] | - | - | - | - | - | - | - | $N^{-\frac{1}{2}}+N^{\frac{-3s}{2s+d}}$ |
| Nguyen et al. [28] | $N^{-\frac{1}{2}}$ | - | - | - | - | - | - | - |
| Moon and Hero [26] | $N^{-\frac{1}{2}}$ | - | $N^{-\frac{1}{2}}$ | $N^{-\frac{1}{2}}$ | $N^{-\frac{1}{2}}$ | $N^{-\frac{1}{2}}$ | $N^{-\frac{1}{2}}$ | $N^{-\frac{1}{2}}$ |

**Assumptions:**  [22]: Both densities $p$ and $q$ must belong to the Hölder class of smoothness $s$, be supported on $[0, 1]^d$
and satisfy $0 < \eta_1 < p, q < \eta_2 < \infty$ on the support for known constants $\eta_1, \eta_2$. [28]: The density ratio $p/q$ must satisfy
$0 < \eta_1 < p/q < \eta_2 < \infty$ and belong to a function class $G$ whose *bracketing entropy* (a measure of the complexity of a
function class) is properly bounded. The condition on the bracketing entropy is quite strong and ensures that the density
ratio is well behaved. [26]: This estimator makes **strong assumptions** to avoid non-parametric rates. Both $p$ and $q$ must
have the same bounded support and satisfy $0 < \eta_1 < p, q < \eta_2 < \infty$ on the support. $p$ and $q$ must have *continuous
bounded* derivatives of order $d$ (which is stronger than assumptions of [22]). $f$ has derivatives of order at least $d$.

# References

[22] A. Krishnamurthy, A. Kandasamy, B. Póczos, and L. Wasserman. Nonparametric estimation of Rényi divergence
and friends. In ICML, 2014.

[26] K. Moon and A. Hero. Ensemble estimation of multivariate f-divergence. In 2014 IEEE International 360
Symposium on Information Theory, pages 356–360, 2014.

[28] XuanLong Nguyen, Martin J. Wainwright, and Michael I. Jordan. Estimating divergence functionals and 363 the
likelihood ratio by convex risk minimization. IEEE Trans. Information Theory, 56(11):5847–5861, 364 2010.


[Meta-Review · NeurIPS 2019]

The paper consider the estimation of f-divergences (of the form \int f(dQ/dP) dP = \int f(q/p) p dz, for convex f) in situations where the density p(z) is known, but q(z) is partially known, i.e., is a mixture of the form q(z) = \int q(Z|x)w(x)dx for known components q(Z|x), but unknown weight distribution W (which however can be sampled from). The main message is that (1) these situations arise naturally in the practice of deep-NN, and (2) that the rates of estimation are parametric in this situation, rather than the worst-case nonparametric rates (depending exponentially on the dimension of Z). The estimator simply substitutes q(z) for a corresponding mixture \hat q(z) defined over a sample {X_i}_1^n from the weight distribution W. The results are non-trivial and consider various conditions relating Q(Z|x) , Q(Z), and P(Z), and most common instances of f-divergences. It however leaves unclear whether these rates are tight in these situations (besides perhaps in a few cases where it matches the best possible parametric rates, albeit the meaning of 'sample size' is different from traditional where Z is being sampled). The main concern about the work, raised by some reviewers, is the assumption that the densities are known or almost known. In other words, the conditions at present seem limited to the NN applications described and leave a sense of narrowness of its potential appeal. However this might not be a problem given the popularity of everything NN.